# Direct bandgap quantum wells in hexagonal Silicon Germanium

Wouter H. J. Peeters [1,6], Victor T. van Lange [1,6], Abderrezak Belabbes[2,3,6], Max C. van Hemert [1], Marvin Marco Jansen [1], Riccardo Farina [1], Marvin A. J. van Tilburg [1], Marcel A. Verheijen [1,4], Silvana Botti [3,5], Friedhelm Bechstedt [3], Jos. E. M. Haverkort [1] & Erik P. A. M. Bakkers [1] ✉

Silicon is indisputably the most advanced material for scalable electronics, but it is a poor choice as a light source for photonic applications, due to its indirect band gap. The recently developed hexagonal $Si_{1-x}Ge_x$ semiconductor features a direct bandgap at least for $x > 0.65$, and the realization of quantum heterostructures would unlock new opportunities for advanced optoelectronic devices based on the SiGe system. Here, we demonstrate the synthesis and characterization of direct bandgap quantum wells realized in the hexagonal $Si_{1-x}Ge_x$ system. Photoluminescence experiments on hex-Ge/$Si_{0.2}Ge_{0.8}$ quantum wells demonstrate quantum confinement in the hex-Ge segment with type-I band alignment, showing light emission up to room temperature. Moreover, the tuning range of the quantum well emission energy can be extended using hexagonal $Si_{1-x}Ge_x$/$Si_{1-y}Ge_y$ quantum wells with additional Si in the well. These experimental findings are supported with ab initio bandstructure calculations. A direct bandgap with type-I band alignment is pivotal for the development of novel low-dimensional light emitting devices based on hexagonal $Si_{1-x}Ge_x$ alloys, which have been out of reach for this material system until now.

Electronic devices based on silicon have been the driver for the revolution in information technology witnessed today. However, with their standard cubic-diamond crystal structure, silicon, germanium, and SiGe-alloys are all indirect band gap semiconductors, impeding the use of silicon-based materials for lasers and optical amplifiers for integrated photonics[1]. Several strategies have been investigated for integrating light emitting materials on silicon, including III-V[2,3], GeSn[4–9], strained Ge[7,10], and SiGe quantum wells and dots[11–17], but remain challenging due to various reasons. When transformed into the hexagonal crystal structure, the hex-$Si_{1-x}Ge_x$ alloys[18] are direct bandgap semiconductors with the fundamental bandgap at the Γ-point. The hex-$Si_{1-x}Ge_x$ compositional family shows tunable light emission from 1.8 μm to 3.4 μm and features a nanosecond radiative lifetime[18]. As such, hex-$Si_{1-x}Ge_x$ stands out in the field of group IV photonics as a direct bandgap semiconductor with a relatively large energy difference between the direct and indirect conduction band minima, up to 0.3 eV for hex-Ge[19,20]. Additional favorable properties of hex-$Si_{1-x}Ge_x$ include its low surface recombination velocity[21], large theoretical Landé g-factor of 18[22], and the potential to fabricate structures from nuclear spin-free isotopes[23], which is important for applications in quantum information.

[1]Department of Applied Physics, Eindhoven University of Technology, 5600 MB Eindhoven, The Netherlands. [2]Department of Physics, Sultan Qaboos University, P.O. Box 123, Muscat, Oman. [3]Institut für Festkörpertheorie und -optik, Friedrich-Schiller-Universität Jena, Jena, Germany. [4]Eurofins Materials Science Netherlands BV, Eindhoven, The Netherlands. [5]Research Center Future Energy Materials and Systems of the University Alliance Ruhr and Interdisciplinary Centre for Advanced Materials Simulation, Ruhr University Bochum, Universitätsstraße 150, Bochum, Germany. [6]These authors contributed equally: Wouter H. J. Peeters, Victor T. van Lange, Abderrezak Belabbes. ✉e-mail: e.p.a.m.bakkers@tue.nl

Quantum confinement in direct bandgap semiconductors has stood at the cradle of many photonic devices such as single photon quantum dot (QD) emitters[24–27], quantum well (QW) lasers[28,29] and colloidal QD LED display technology[30–32]. These direct bandgap low dimensional structures have been responsible for major advances in science and constitute a toolbox for many optoelectronic and quantum photonic devices[33,34], allowing for tunable and narrow band emission, and the concentration of charge carriers.

Here, we show the synthesis of hex-SiGe quantum wells, and we demonstrate quantization of the energy levels with type-I band alignment between the hex-$Si_{1-x}Ge_x$ well ($0.9 < x < 1.0$) and the hex-$Si_{1-y}Ge_y$ barrier ($0.7 < y < 0.8$). We observe broad tunability of the QW emission from 3.4 µm for hex-$Ge/Si_{0.2}Ge_{0.8}$ to 2.0 µm for hex-$Si_{0.1}Ge_{0.9}$/$Si_{0.3}Ge_{0.7}$, which may be further extended down towards 1.5 µm, the limits of which are a subject of future investigations. Most notably, we confirm direct bandgap emission from the QWs by observing a subnanosecond photoluminescence lifetime, comparable with direct bandgap emission in bulk hex-SiGe. Our experimental data are complemented by ab initio density functional theory and quasiparticle calculations of the bandstructure of hex-$Ge/Si_{0.25}Ge_{0.75}$ QWs, showing a direct bandgap with a large directness, defined to be the separation between the $\bar{\Gamma}$ minimum and the nearest indirect conduction band minimum. Theory confirms a type-I heterostructure and carrier confinement in the hex-Ge layers, with almost identical valence and conduction band offsets. Our hex-$Ge/Si_{0.2}Ge_{0.8}$ QWs thus can serve as a textbook example demonstrating quantum confinement.

## Results

### Growth and structural analysis of hex-$Ge/Si_{0.2}Ge_{0.8}$ QWs

We have embedded coaxial hex-Ge quantum wells in hex-$Si_{0.2}Ge_{0.8}$ barriers, grown epitaxially on the $\{1\bar{1}00\}$ m-plane facets of wurtzite (WZ) GaAs core nanowires (NWs)[18,35], as shown in Fig. 1a. The goal is to create a QW of hex-Ge, as shown in Fig. 1b. A Scanning Electron Microscopy (SEM) image in Fig. 1c illustrates the dimensions of the resulting structures. The $Ge/Si_{0.2}Ge_{0.8}$ shells in these NWs are doped with arsenic, at a doping level below $2.5 \times 10^{18}$ cm$^{-3}$ (See "Methods" for details about the growth).

The $Ge/Si_{0.2}Ge_{0.8}$ QWs are characterized by cross-sectional Scanning Transmission Electron Microscopy (STEM) along two different zone axes. When imaged along the [0001] zone axis, the $Ge/Si_{0.2}Ge_{0.8}$ QW is visible as a hexagon, an example is given in Fig. 2a, and other data is shown in Fig. S2. We note that the $Si_{0.2}Ge_{0.8}$ barrier has composition fluctuations, Si-rich spokes connect the corners of the GaAs

with the outer corners of the NW[36]. Moreover, as highlighted in the inset of Fig. 2a, the thickness of the Ge QW varies between the different facets. Fluctuations in QW thickness on different facets have also been reported for other material systems[37,38], possibly resulting in charge carrier localization in the thickest well[39]. The QW thickness varies between 10 and 30 nm by changing the growth time, as shown in Fig. 2b, while the $Si_{0.2}Ge_{0.8}$ barrier thickness always exceeds 50 nm. For each sample, we observe a distribution of thicknesses, mainly due to the facet-to-facet fluctuation within one NW, which is larger than the deviation in average QW thickness between different NWs of the same sample. The probability distribution is bimodal for some samples, with two different Ge QW thicknesses that are most likely. However, the bimodal distribution does not appear for all samples, and therefore the average is taken as a measure of the QW thickness.

When imaged along the [11$\bar{2}$0] zone axis, the $Ge/Si_{0.2}Ge_{0.8}$ QW is visible as a vertical stripe in TEM (Fig. 2c). The thickness of the QW is not constant along the length of the NW (Fig. S3), and the roughness on the $\{1\bar{1}00\}$ interface between $Ge/Si_{0.2}Ge_{0.8}$ is estimated from Fig. 2c to be a few nm. Additionally, the [11$\bar{2}$0] zone axis allows to distinguish between hexagonal and cubic stacking. The hexagonal stacking is not continuous along the [0001] direction but is segmented due to the inclusion of cubic defects. Most of these are I3 defects, which nucleate either on the GaAs-$Si_{0.2}Ge_{0.8}$ interface or at random positions in the shell[40,41]. An example is indicated with the arrow in Fig. 2c. A statistical analysis of the atomic stacking shows a broad distribution in the length of segments with the hexagonal stacking (Fig. S4a, b). In contrast, only narrow segments of coherent cubic stacking are observed.

X-ray diffraction (XRD) is used to study the crystalline quality and lattice constants from a large ensemble of NWs. The diffraction spectra of all samples are similar, indicating comparable crystalline quality between samples (Fig. S4c, d). A reciprocal space map around the hexagonal [10$\bar{1}$5] reflection shows a single peak (Fig. 2d), despite the 0.8% lattice mismatch between Ge and $Si_{0.2}Ge_{0.8}$. Increasing the Ge thickness does not significantly influence the lattice parameters of the NWs (Fig. S5a). Instead, the c-lattice constant depends on the thickness of the $Si_{0.2}Ge_{0.8}$ barriers (Fig. S5b). These observations indicate that there is pseudomorphic strain relaxation in the $Ge/Si_{0.2}Ge_{0.8}$ structures.

The $Si_{0.2}Ge_{0.8}$ barriers have smaller lattice constants than the Ge QW, and the Ge is therefore compressed along the $\langle 11\bar{2}0 \rangle$ and $\langle 0001 \rangle$ directions. Pseudomorphic strain relaxation in the Ge QW results in an increased lattice constant along the $\langle 1\bar{1}00 \rangle$ direction. This radial relaxation becomes more pronounced if the Ge thickness is increased, as confirmed by the Geometric Phase Analysis (GPA) of TEM images (Fig. S5c, d).

### Photoluminescence of hex-$Ge/Si_{0.2}Ge_{0.8}$ QWs

The optical properties of the $Ge/Si_{0.2}Ge_{0.8}$ QW samples have been studied by low-temperature photoluminescence (PL) as a function of the QW thickness in Fig. 3a. We observe that the emission energy consistently blueshifts with decreasing QW growth time demonstrating increasing quantum confinement with decreasing thickness. Moreover, all QW emission peaks are positioned between the emission originating from the bulk hex-Ge and hex-$Si_{0.2}Ge_{0.8}$ reference samples, thus providing experimental evidence for type-I band alignment. We note that for type-II band alignment, one would expect emission below the energy of (strained) bulk hex-Ge[42]. The width of the QW emission peaks is larger than that of the reference samples, and for some samples, multiple peaks have been observed; this is probably due to fluctuations in QW thickness and, for the wider QWs, the presence of the second confined level. The intensity of the QW emission exceeds that of the reference sample (see Fig. S6a), indicating that many carriers diffuse towards the QWs. The relation between emission energy and QW thickness is shown in Fig. 3b, showing a blueshift with decreasing thickness, consistent with a shift due to confinement energy in a QW.

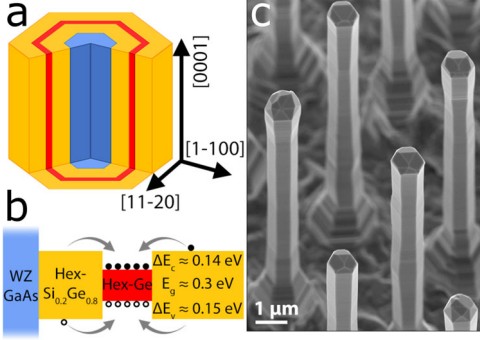

**Fig. 1 | Quantum wells of hex-$Ge/Si_{0.2}Ge_{0.8}$. a** Schematic illustration of the GaAs/$Si_{0.2}Ge_{0.8}$/Ge/$Si_{0.2}Ge_{0.8}$ core-multishell nanowires. All interfaces are orthogonal to $\langle 1\bar{1}00 \rangle$ directions. **b** Schematic band alignment of the different materials. The electrons and holes are confined in the hex-Ge layer due to type-I alignment with the surrounding hex-$Si_{0.2}Ge_{0.8}$, as will be proven in this manuscript. Approximate values of the bandgap and offsets are given. **c** 30-degree tilted scanning electron micrograph of a NW array. Within these NWs, a (12 ± 3) nm $Ge/Si_{0.2}Ge_{0.8}$ QW is embedded.

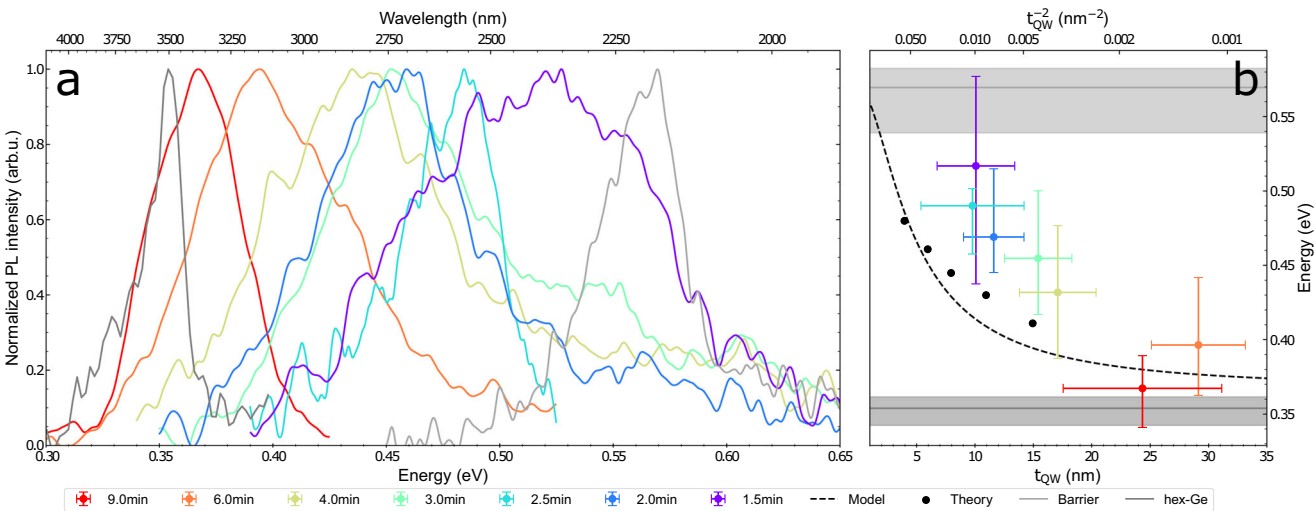

**Fig. 2 | Structural properties of the studied Ge/Si$_{0.2}$Ge$_{0.8}$ QWs. a** False-colored HAADF-STEM image of a cross-sectional lamella, viewing the Ge QW along the [0001] zone axis. Inset shows that Ge QWs on neighboring facets have different thicknesses. **b** Growth rate curve for Ge/Si$_{0.2}$Ge$_{0.8}$ QWs. The thicknesses of individual facets, all measured in images acquired along the [0001] zone axis, are indicated with the colored data points. Colored areas show approximate probability distributions, obtained from these data points by Kernel smoothing. **c** False-

colored HAADF-STEM image of a cross-sectional lamella, viewing the QW along the [11$\bar{2}$0] zone axis. The core of the NW is on the left. Locations with local hexagonal (ABABA, blue), cubic (ABCA, green), and twinned cubic boundary (ABCBA, pink) stacking are indicated with circles. The pink arrow highlights a defect that starts in the Ge QW. **d** X-ray diffraction reciprocal space map around the hexagonal [10$\bar{1}$5] reflection. The peak position does not match Vegard's rule (dashed line), indicating pseudomorphic strain relaxation.

**Fig. 3 | Quantum confinement in hex-Ge/Si$_{0.2}$Ge$_{0.8}$ QWs. a** Ge/Si$_{0.2}$Ge$_{0.8}$ PL spectrum for varying growth time at low temperature ($T \approx 4$ K) and low excitation density ($P \leq 65$ W cm$^{-2}$), **b** The PL emission versus the QW thicknesses $t_{QW}$ determined from TEM, together with the confinement energy predicted from theory shifted up by 60 meV to account for the difference in the theoretical and experimental bandgap of the hex-Ge. The dashed line shows the confinement energies

using a simple finite QW model. We also include the reference spectra of bulk-Ge and the bulk Si$_{0.2}$Ge$_{0.8}$ barrier as horizontal lines with the FWHM of the spectra shown as horizontal gray bars. Error bars in $t_{QW}$ are the standard deviations presented in Fig. 2b and error bars in the peak energy indicate the FWHM of the emission spectrum.

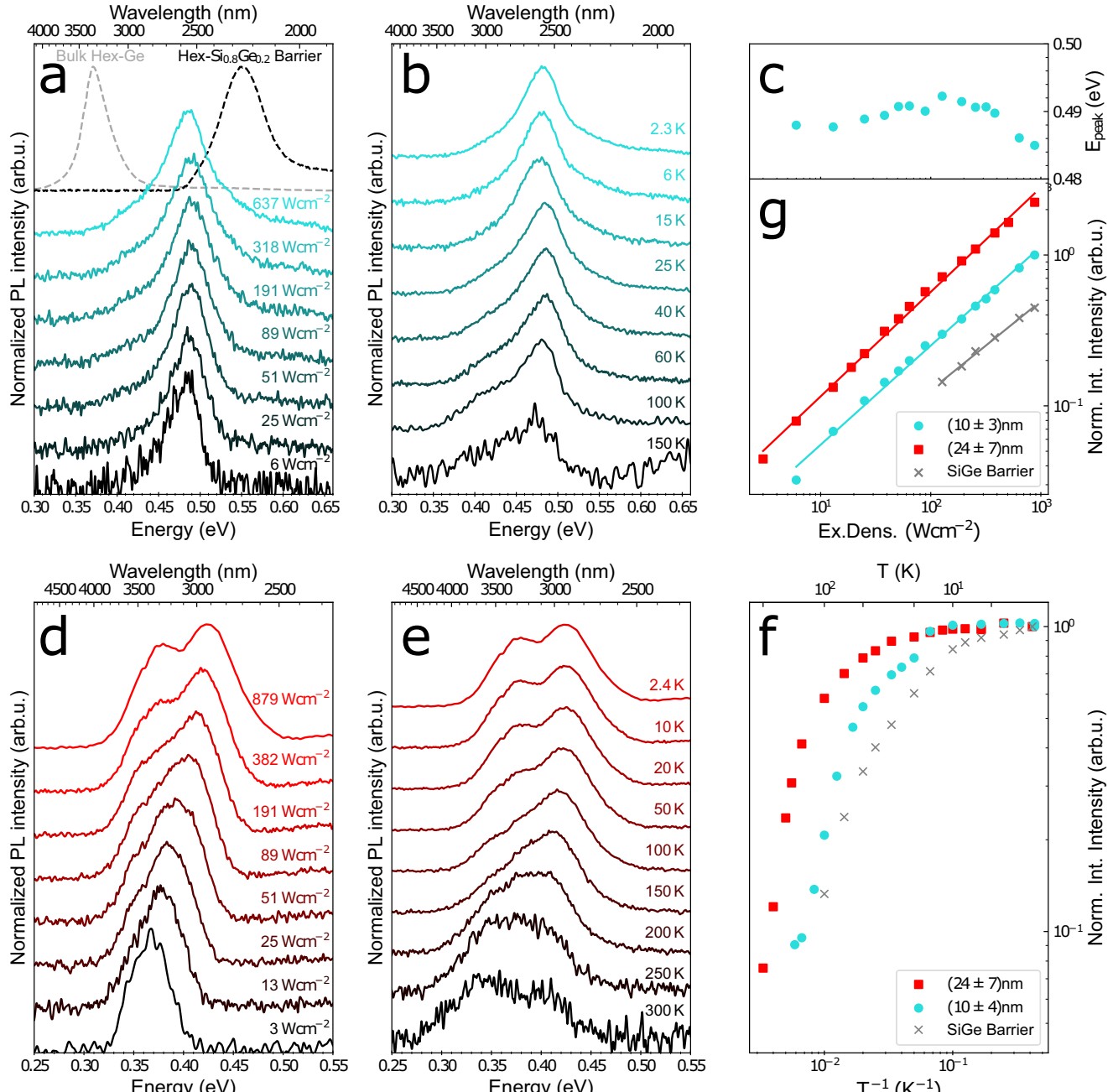

**Fig. 4 | Type-I band alignment in hex-Ge/$Si_{0.2}Ge_{0.8}$ QWs. a** The $(10 \pm 4)$ nm (2.5 min) low temperature ($T \approx 4$ K) QW photoluminescence spectrum as a function of excitation density showing a constant lineshape over two orders of magnitude with the peak position in between the bulk-Ge and $Si_{0.2}Ge_{0.8}$ barrier reference measurements, **b** The $(10 \pm 4)$ nm QW showing a near constant lineshape through temperature with the tail states becoming slightly more significant as the peak intensity quenches at higher temperatures. **c** The emission peak energy of the $(10 \pm 4)$ nm QW shows a nearly constant magnitude through excitation density. Initially the peak blueshifts due to band-filling of the QW and then redshift around 100 W cm$^{-2}$, likely due to Bandgap renormalization. **d** The $(24 \pm 7)$ nm (9 min) QW spectrum evolves from a single to a double peak with increasing excitation density

due to band-filling. Additionally, if the lowest and highest excitation density spectra are compared, we observe no significant shift in the position of the low energy peak. **e** The $(24 \pm 7)$ nm QW sample as a function of temperature showing emission up to room temperature. **f** The Arrhenius plot of the QWs and $Si_{0.2}Ge_{0.8}$ barrier reference samples measured at an excitation density of 0.88 kW cm$^{-2}$. It can be seen that the temperature behavior of the QWs exceeds the bulk hex-$Si_{0.2}Ge_{0.8}$ reference. **g** The Light-In Light-Out (LILO) curves of the QWs and SiGe barrier reference samples measured at 4 K. The slopes of $(0.69 \pm 0.01)$ and $(0.66 \pm 0.01)$ for the $(24 \pm 7)$ nm and $(10 \pm 4)$ nm QWs respectively exceed the $(0.59 \pm 0.02)$ of the bulk hex-$Si_{0.2}Ge_{0.8}$ reference.

The optoelectronic properties of the Ge/$Si_{0.2}Ge_{0.8}$ QWs are investigated in more detail by power- and temperature-dependent photoluminescence spectroscopy. We focus here on two specific samples: (i) a relatively thin $(10 \pm 4)$ nm QW showing single peak emission with strong confinement and (ii) a thick $(24 \pm 7)$ nm QW with small confinement energy and a large separation between the

confinement level in the QW and the barrier, as shown in Fig. 4. Besides the emission being between the hex-Ge reference and the $Si_{0.2}Ge_{0.8}$ barrier, as mentioned before, we observe that the emission peak energy of the $(10 \pm 4)$ nm QW is nearly independent of both the excitation density in Fig. 4a and the temperature in Fig. 4b. We highlight the (absence of) shift with excitation density in Fig. 4c. At low

excitation densities a minor < 5 meV blueshift is observed, followed by a redshift at high excitation. These shifts are likely due to Burstein-Moss band-filling (blueshift) and bandgap renormalization (redshift). Importantly, we do not observe the significant blueshift with increasing excitation density expected for a type-II QW structure[43]. The absence of such a blueshift provides additional evidence for a type-I band offset. Similar trends have been observed for the other QW samples. The spectra of the thick (24 ± 7) nm QW sample are plotted in Fig. 4d as a function of excitation density. At low excitation density, we observe a single emission peak, while with increasing excitation density, the sample evolves from a single to a double peak shape. We attribute the presence of the second peak at increased excitation density to either distinct QW thicknesses e.g., at different facets of the nanowire shells or to the observation of the HH2-C2 transition within the wide quantum well. The behavior of the high energy peak becomes dominant at intermediate excitation densities, while the lower energy peak increases at the highest excitation densities. This could indicate a different density of states of the subbands[44], but a detailed analysis is beyond the scope of the present paper. The light-in light-out (LILO) curves for the QWs and the $Si_{0.2}Ge_{0.8}$ barrier reference sample are introduced in Fig. 4g. While we observe sublinear behavior, the slopes of (0.69 ± 0.01) and (0.66 ± 0.01) for the (24 ± 7) nm and (10 ± 4) nm QWs respectively exceed the slope of the barrier reference sample (0.59 ± 0.02) (observed for all QW samples shown in Fig. S6b). Pure radiative (non-radiative) recombination is expected to yield a slope of 1 (2). A LILO slope below unity is due to an increasing loss of carriers at high excitation, which is most likely due to carrier overflow into cubic insertions, or due to Auger recombination. This behavior deserves further study.

We present the PL as a function of temperature in Fig. 4e. Notably, room temperature emission from an ensemble of NWs with a *single* coaxial hex-Ge/$Si_{0.2}Ge_{0.8}$ QW is demonstrated. In the range $T = 2.4–100$ K, the relative magnitude of the higher energy peak increases, which is likely due to the de-trapping of carriers from the potential landscape due to alloy fluctuations in the $Si_{0.2}Ge_{0.8}$ barrier, allowing more carriers to diffuse to the QW, while the lower energy QW level is already fully occupied. Above 250 K the low energy peak again becomes more dominant, which is likely due to a higher probability of thermal emission from the higher energy QW level into the barrier, while also allowing the carriers to be even more mobile to find the

lowest energy states. The temperature dependence of the integrated PL intensity is shown in Fig. 4f and shows a monotonous decay of the intensity with temperature. This shows that the emission is not pho-non-activated, which is a strong indication for direct bandgap emission[18]. Moreover, the intensity of the QW emission outperforms the emission of the bulk hex-$Si_{0.2}Ge_{0.8}$ reference sample at elevated temperatures (observed for all QW samples shown in Fig. S6c), which is an important advantage for devices e.g., a hex-Ge/$Si_{0.2}Ge_{0.8}$ QW laser. From the thermal quenching results we estimate the band offset and effective mass of the most shallow confined charge carrier from the activation energies in Fig. S6d of three of the widest (approximately infinite) QW samples which are found to be $E_{offset} = (100 ± 30)$meV and $m^* = (0.03 ± 0.02)m_0$ respectively, which is close to the predicted band offset and effective mass of our ab initio bandstructure calculations presented below.

## $Si_{1−x}Ge_x$/$Si_{1−y}Ge_y$ Alloy/Alloy QWs

Having confirmed quantum confinement and wavelength tunability of emission from the hex-Ge/$Si_{0.2}Ge_{0.8}$ QWs, we subsequently like to demonstrate type-I confinement in hex-$Si_{0.1}Ge_{0.9}$/$Si_{0.3}Ge_{0.7}$ QWs that emit light at even higher energy by making use of alloys with a larger bandgap[18]. These hex-$Si_{0.1}Ge_{0.9}$/$Si_{0.3}Ge_{0.7}$ QWs are realized as coaxial nanowire shells, similar to those presented in Fig. 1. A cross-sectional view of the (5 ± 1) nm $Si_{0.1}Ge_{0.9}$/$Si_{0.3}Ge_{0.7}$ QW is presented in Fig. 5a, and an overview of all studied $Si_{0.1}Ge_{0.9}$/$Si_{0.3}Ge_{0.7}$ QWs is presented in Fig. S7. There are two main differences compared to the Ge/$Si_{0.2}Ge_{0.8}$ system studied. Additional radial contrast lines, which do not terminate at the NW corners, are recognizable in the TEM image. These lines correspond to dislocations, whose occurrence is correlated with the lattice mismatch between the WZ GaAs core and the hex-$Si_{1−x}Ge_x$ shell. Secondly, there is a compositional gradient in the $Si_{1−x}Ge_x$ barrier, where the Si concentration increases with increasing distance to the GaAs core (see Fig. S8). Both effects arise from the lattice mismatch in this system, which is either relaxed through dislocations or mitigated by forming a self-assembled compositional gradient buffer layer.

The photoluminescence emission from the hex-$Si_{0.1}Ge_{0.9}$/$Si_{0.3}Ge_{0.7}$ QW is between the emission of the bulk $Si_{0.1}Ge_{0.9}$ well material, and the barrier material, as shown in (Fig. 5b), signifying a type-I band offset also for these compositions. We again fit the observed QW emission energies with the conventional finite QW

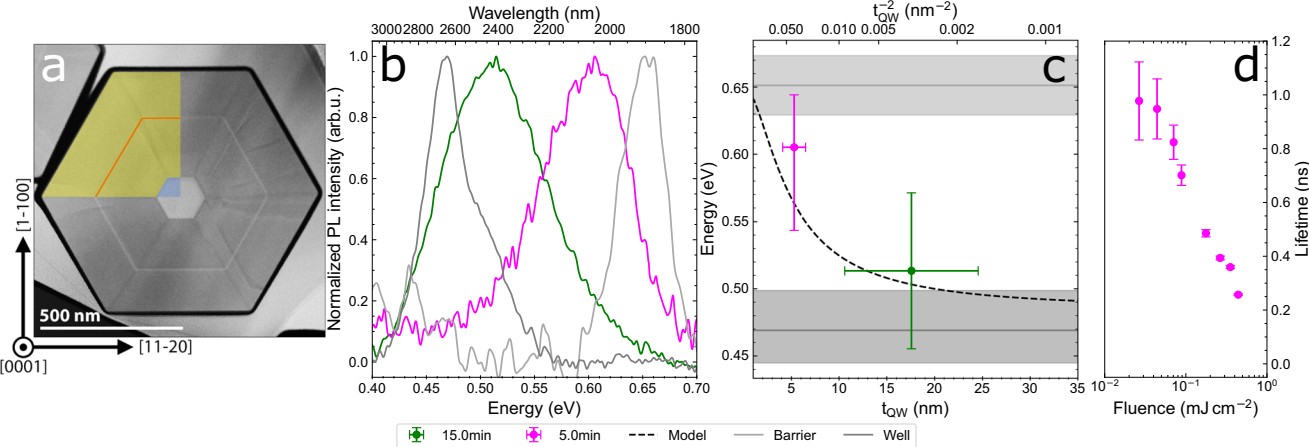

**Fig. 5 | Studies of hex-$Si_{0.1}Ge_{0.9}$/$Si_{0.3}Ge_{0.7}$ QWs. a** False-colored HAADF-STEM of a cross-sectional lamella, viewing the (5 ± 1) nm (5 min) $Si_{0.1}Ge_{0.9}$/$Si_{0.3}Ge_{0.7}$ QW in the [0001] zone axis. **b** Background corrected photoluminescence spectra for varying QW growth time at low temperature (≈ 4 K) and high excitation density < 0.88 kW cm⁻². Reference spectra of bulk $Si_{0.1}Ge_{0.9}$ and $Si_{0.3}Ge_{0.7}$ are included. **c** The PL emission versus the QW thicknesses $t_{QW}$ determined from TEM. Spectra of the $Si_{0.1}Ge_{0.9}$ well and $Si_{0.3}Ge_{0.7}$ barrier alloys are included as horizontal lines with the FWHM of the

spectra as horizontal gray bars. A simple finite QW model is calculated for this heterostructure which shows reasonable agreement with the experiment. Error bars in $t_{QW}$ are the standard deviations presented in Fig. S7e and error bars in the peak energy indicate the FWHM of the emission spectrum. **d** Initial QW lifetime measured using TCSPC for the (5 ± 1) nm QW for varying laser fluence with the error bars indicating the standard deviation determined fitting the initial decays presented in Fig. S9b.

model, showing qualitative agreement in Fig. 5c. This suggests that the band alignment of the broader family of the hex-$Si_{1-x}Ge_x$/$Si_{1-y}Ge_y$ QWs is of type-I nature.

We emphasize that the observation of efficient direct bandgap emission is not obvious since theoretical DFT calculations predict[18] a radiative lifetime of 20 μs for hex-Ge. If true, this would comprise the well material of our hex-Ge/SiGe QWs. To obtain experimental evidence for direct bandgap emission, we measure the carrier recombination lifetime using a Time-Correlated Single Photon Counting (TCSPC) system employing a Superconducting Nanowire Single Photon Detector (SNSPD) for the $(5 \pm 1)$ nm QW (Single nanowire spectrum shown in Fig. S9a). We measure the PL lifetime at a lattice temperature of 4 K where the nonradiative recombination rate is expected to vanish since the nonradiative recombination is a thermally activated process by $\tau_{nr}^{-1} = \tau_{nr}^{-1} e^{-E_a/kT}$. For our QWs, this behavior is experimentally observed as a constant PL-intensity below a temperature of 10 K and at an excitation density of 0.88 kW cm$^{-2}$ in Fig. 4f. We measure the PL decay time under pulsed excitation conditions where the radiative limit is maintained up to much higher temperature as shown by Fadaly et al.[18], implying that the measured PL decay time should be equal to the radiative lifetime at 4 K. We present the carrier recombination lifetime in Fig. 5d for varying laser fluence. Importantly, we observe an initial carrier lifetime of ≈ 1 ns for the lowest fluence (Full time decays are provided in Fig. S9b), confirming direct bandgap emission. We note that the observation of a decreasing recombination lifetime with increasing excitation density provides additional evidence for radiative recombination governed by $1/\tau_{rad} = B(n_0 + \Delta n)(p_0 + \Delta p)/\Delta p \approx B\Delta n$ for high excitation ($\Delta n = \Delta p >> n_0, p_0$), in which $B$ is the coefficient for radiative recombination, $n_0, p_0$ are the doping concentrations and $\Delta n, \Delta p$ are the photoexcited carrier concentrations. On the other hand, the observations in Fig. 5d cannot be explained by a nonradiative recombination mechanism since nonradiative recombination centers get saturated at high excitation, thus increasing the lifetime. We conclude that the observed nanosecond radiative recombination lifetime falls within the same range as that reported by Fadaly et al.[18,45] for bulk hex-SiGe nanowires and confirms direct bandgap emission in $Si_{0.1}Ge_{0.9}$/$Si_{0.3}Ge_{0.7}$ QWs.

## Ab initio calculations

To examine the band alignment of the experimentally realized hex-Ge/$Si_{0.2}Ge_{0.8}$ and $Si_{0.1}Ge_{0.9}$/$Si_{0.3}Ge_{0.7}$ single QWs, we first calculate the electronic band structure of hex-Ge/$Si_{0.25}Ge_{0.75}$ multi-quantum well (MQW) structures, with $(1\bar{1}00)$ interfaces, as superlattices (see Fig. 6a). The ab initio calculations are based on Density Functional Theory (DFT) for optimized atomic geometries and an approximate quasiparticle (QP) electronic structure approach to the band structures (see "Methods" for details). The band structures of the different materials and heterostructures are aligned employing their branch points (BPs)[46]. The Ge/$Si_{0.25}Ge_{0.75}$ MQW system is the closest approximation of the experimentally realized Ge/$Si_{0.2}Ge_{0.8}$ QWs, which still allows modeling of the alloy barriers by ordered arrangements of a single Si and three Ge atoms in one Lonsdaleite unit cell. The increase of the average Si incorporation by 5% compared to the experiment increases the barrier heights by approximately 0.05 eV, but has a vanishing effect on the confinement for both carrier types. Within the calculations, the $Si_{0.25}Ge_{0.75}$ barrier thickness is kept constant at 2 nm, i.e., 12 monolayers along the $[1\bar{1}00]$ direction, while the Ge well thickness is varied between 4 and 15 nm. This barrier thickness is sufficient to prevent tunneling of electron and hole wave functions through the barriers[47]. As a consequence, the Ge layers in the MQW system are electronically decoupled, and the Ge layers can thus be treated as isolated single QWs. The use of thin $Si_{0.25}Ge_{0.75}$ barriers in the modeling only affects the strain distribution, which is different for the thick $Si_{0.2}Ge_{0.8}$ barriers in the experiment. This effect is accounted for by applying an external biaxial strain to the Ge/$Si_{0.25}Ge_{0.75}$ MQW structure of -0.6%

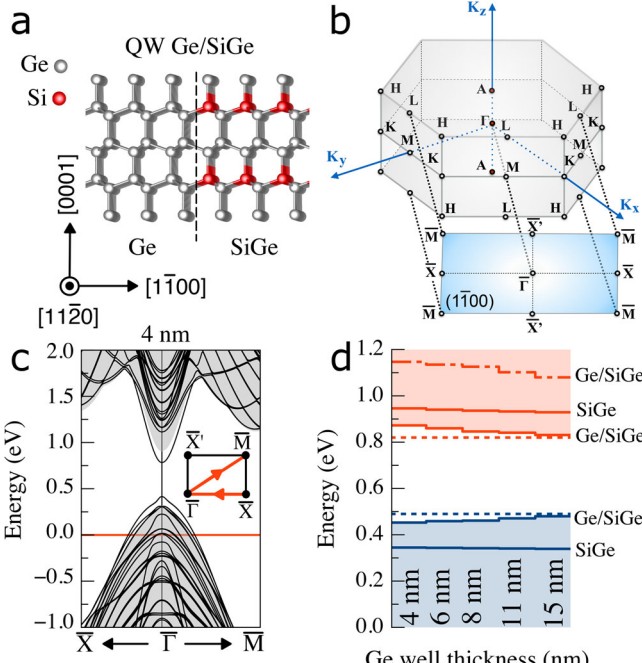

**Fig. 6 | Band structure calculations of hex-Ge/$Si_{0.25}Ge_{0.75}$. a** Hexagonal Ge/$Si_{0.25}Ge_{0.75}$ heterostructure with $(1\bar{1}00)$ interfaces. **b** Bulk hexagonal Brillouin zone (BZ) and its projection onto the two-dimensional BZ of the $(1\bar{1}00)$ interface. **c** Direct bandgap band structure of hexagonal 4 nm Ge/ 2 nm $Si_{0.25}Ge_{0.75}$ multiple quantum well structure (black lines) and bulk $Si_{0.25}Ge_{0.75}$ (gray area) projected onto the two-dimensional Brillouin zone. The horizontal red line indicates the branching points of the two systems used as energy zero for alignment. **d** Energies of the lowest electron and highest hole subband at $\bar{\Gamma}$ versus Ge thickness in the Ge/$Si_{0.25}Ge_{0.75}$ heterostructures studied. They are compared with the lowest conduction and highest valence band of the bulk $Si_{0.25}Ge_{0.75}$ barrier material, see the "Methods" section for an explanation. Dashed lines indicate the extrapolated band-states at infinite Ge well thickness. For comparison, also the energy position of the lowest indirect conduction band minimum outside $\bar{\Gamma}$ (dot-dashed line) is given.

and -0.91% along the $[11\bar{2}0]$ and $[0001]$ directions respectively, based on the X-ray diffraction experiments on the realized Ge/$Si_{0.2}Ge_{0.8}$ QWs (Fig. S5a). The studied heterostructure is allowed to relax along the $[1\bar{1}00]$ direction, tending towards an ≈0.3% expansion in the well and ≈0.1% contraction in the $Si_{0.25}Ge_{0.75}$ barrier.

The QP band structure of a (superlattice with a) 4 nm thick Ge layer is displayed in Fig. 6b, c, clearly showing a direct bandgap with a $\bar{\Gamma}$ minimum approximately 0.3 eV below the lowest indirect conduction band minimum which appears near the corner point $\overline{M}$ of the Brillouin zone boundary. We plot the band structure of the MQW together with a background illustrating the projected band structure of the strained $Si_{0.25}Ge_{0.75}$ bulk. The two band structures are aligned by their BPs. The bands of the Ge/$Si_{0.25}Ge_{0.75}$ MQW, within the fundamental gap of the projected $Si_{0.25}Ge_{0.75}$ band structure, describe subbands of electrons and holes, whose wave functions are both localized in the Ge layers. The localization of both the electron and hole wave functions in the Ge well (Fig. S10a) clearly indicates type-I band alignment. The type-I behavior is confirmed by the energies for the highest hole subbands and lowest-energy electron subbands at the $\bar{\Gamma}$ point, which are presented versus the Ge layer thickness in Fig. 6d. Corresponding band structures for MQW structures with thicker Ge layers are displayed in Fig. S10b. Combining this data with the calculated band structure for bulk (strained) hex-Ge, serving as infinitely thick QW, allows us to extract the quantization effects, more precisely the confinement energies of the lowest $n = 1$ electron and hole levels directly from ab initio band structure calculations. While only one level appears in the

narrow QW with a thickness of 4 nm, a second and third confined level appear in finite QWs starting from a thickness of 8 nm (Fig. S10b).

The band offsets in the conduction band and the valence band of 0.13–0.15 eV are nearly equal (Fig. S10c). The band offsets can be employed as barrier heights in simplified rectangular finite QW models for electrons and holes. The ab initio confinement energy of electrons (holes) in the QW vary from 72 (36) to 31 (8) meV for thicknesses of 4 and 15 nm, respectively. These values are much smaller than the offsets, and one therefore may approximate the system as an infinite QW. For the lowest $n = 1$ levels, the finite band offsets $\Delta E_{c/h}$, and the mentioned ab initio confinement energies $\epsilon_{e/h}$ in the finite rectangular-well model allows the extraction of the effective electron/hole masses according to[48]

$$m^*_{e/h} = \frac{\hbar^2}{2 t^2_{QW} \epsilon_{e/h}} \left( 2 \arctan \sqrt{\frac{\Delta E_{c/v} - \epsilon_{e/h}}{\epsilon_{e/h}}} \right)^2 \qquad (1)$$

as $m_e \approx 0.05\, m_0$ and $m_h \approx 0.13\, m_0$ averaged over all studied QWs. These values are close to those as $0.076\, m_0$ and $0.055\, m_0$ which have been calculated for unstrained bulk hex-Ge along the $[1\bar{1}00]$ direction[19]. Computations without external strain result in much smaller confinement energies, which indeed are closely related to the bulk effective masses of unstrained hex-Ge.

## Discussion

The experimental values for the bandgaps of the 4, 6, 8, 11 and 15 nm Ge/Si$_{0.2}$Ge$_{0.8}$ QWs are compared with the calculated results (black dots) in Fig. 3b. For properly comparing theory with experiment, the theoretical bandgaps are shifted with +60 meV to match the calculated bandgap of the hex-Ge well ($\approx 0.30$ eV)[19] with the experimentally observed bandgap of bulk hex-Ge ($\approx 0.36$ eV)[18]. This shift remains within the error margin of the ab initio DFT calculations ($\approx 0.1$ eV or 25%[49]). Based on the theoretically calculated band offsets and effective masses[19,50,51], the emission energy versus thickness is also calculated using a conventional finite QW model (dashed line)[52]. This simple model is useful to calculate the emission energies for any QW thickness and composition when reasonable values for the band offsets and carrier masses are available and detailed QP calculations are computationally unfeasible.

A qualitative agreement between theory and experiment is obtained, but the experimental emission energies are all higher than the theoretical values. We identify three possible reasons for the deviation between experiment and theory. (1) The Ge QW thicknesses, measured from TEM images, are slightly overestimated (see Fig. S3). (2) We do not include an additional confinement energy due to quantization along the length of the NW, due to the inclusion of cubic stacking faults. The alignment between cubic and hexagonal stacking is expected to be of type-I[47,53], and every hexagonal segment with a direct bandgap is thus bound by cubic barriers with larger bandgap (Fig. S4a). The exact increase of confinement due to the cubic insertions is ambiguous and subject of future investigations. (3) Likely a few percent Si is incorporated in the wells due to interdiffusion of Si between the Si$_{0.2}$Ge$_{0.8}$ and the Ge wells, which elevates their bandgap since the bandgap of hex-Si$_{1-x}$Ge$_x$ alloys is larger than that of hex-Ge[18]. Moreover, interdiffusion of Si results in a less steep potential at the QW-Barrier interface, which might also increase the confinement energies.

In conclusion, we have grown coaxial hex-Ge/Si$_{0.2}$Ge$_{0.8}$ and Si$_{0.1}$Ge$_{0.9}$/Si$_{0.3}$Ge$_{0.7}$ QWs showing direct bandgap light emission. We experimentally confirm efficient direct bandgap emission by the temperature dependence of the integrated PL versus temperature as well as by the observed carrier lifetime of $\approx 1$ ns at 4 K, where the recombination is purely radiative. The direct bandgap is confirmed by ab initio DFT and approximate quasiparticle calculations showing a high directness, implying that the indirect minima are 0.3 eV above the

$\bar{\Gamma}$ minimum. In addition, we observe clear quantum confinement combined with type-I band alignment. Importantly, both analyses of the thermal quenching observed in the Arrhenius pots as the theoretical calculations demonstrate nearly equal conduction and valence band offsets. Although our hex-Ge/Si$_{0.2}$Ge$_{0.8}$ QWs are lattice mismatched and feature strongly anisotropic effective masses, our results can still be properly described by a simple finite QW model. In this paper, we studied hex-Si$_{1-x}$Ge$_x$/Si$_{1-y}$Ge$_y$ nanowire QWs, but our findings are expected to equally apply to future planar hex-Si$_{1-x}$Ge$_x$/Si$_{1-y}$Ge$_y$ QWs compatible with Si-photonics circuits. Our results are unlocking the hex-Si$_{1-x}$Ge$_x$/Si$_{1-y}$Ge$_y$ system for different low-dimensional devices for photonics and quantum information, such as quantum well lasers, optical amplifiers and single photon sources using Si$_{1-x}$Ge$_x$ alloys.

## Methods
### Growth
The nanowires are grown on commercially available GaAs (111)B oriented (n-doped with Si) substrates. The substrates are cleaned with an NH$_4$OH treatment before e-beam exposure, for which an PMMA-A2 photoresist is used. The e-beam exposure patterns 300 x 300 nm squares with a pitch of 4 μm. The resist is developed after e-beam exposure with a solution of MIBK/IPA, followed by an Au deposition of 6 nm by e-beam evaporation. The resist is removed in a lift-off process with PRS3000, acetone and IPA. Final steps before growth involve an oxygen plasma treatment to remove organic residues, followed by an NH$_4$OH treatment to remove surface oxides. The epitaxial growth of WZ GaAs NWs is done in a close-coupled shower head Metal Organic Vapor-Phase Epitaxy (MOVPE) reactor and follows the recipe detailed in Fig. S1a. The total flow through the reactor is 8.2 liters per minute. The obtained wurtzite (WZ) GaAs NWs can have stacking fault densities as low as 4 per μm.

Removal of the Au catalyst is done by wet-chemical etching in a diluted cyanide solution (KCN:H$_2$O - 1:10) for 17 min. KCN residues are removed by rinsing in H$_2$O for 20 min. The rinsing is immediately followed by a NH$_4$OH treatment to remove oxides from the NW sidewalls. The samples are immersed in IPA for 30 seconds, after which the samples are ready to be dried through centrifugation. The samples with GaAs NWs are placed inside the MOVPE reactor to clean the sidefacets of the NWs, according to the recipe detailed in Fig. S1b. The GaAs shell growth during this step is negligible.

The samples are taken out of the reactor, and the reactor kit is changed to a kit dedicated to the growth of Si$_{1-x}$Ge$_x$ alloys. Extensive coating recipes are used to ensure minimal contamination from previous GaAs runs. Earlier work on hex-Si$_{1-x}$Ge$_x$ showed a high As-doping level in the order of $9 \times 10^{18}$ cm$^{-3}$ as deduced by Atom Probe Tomography (APT) measurements[18]. Equivalent samples as those presented in this study still have unintentional doping from arsenic, but the level is reduced below the detection limit of APT, so it is below $2.5 \times 10^{18}$ cm$^{-3}$, likely due to the extensive coating runs.

Samples with WZ GaAs cores are reintroduced in the reactor. Different recipes are used for hex-Ge/Si$_{0.2}$Ge$_{0.8}$ and Si$_{0.1}$Ge$_{0.9}$/Si$_{0.3}$Ge$_{0.7}$ samples, as detailed in Fig. S1c–d. For hex-Ge/Si$_{0.2}$Ge$_{0.8}$ QWs, the first step is an anneal in an H$_2$ atmosphere, which improves the GaAs-Si$_{0.2}$Ge$_{0.8}$ interface. The initial Si$_{0.2}$Ge$_{0.8}$ shell is grown in 90 min. However, a large fraction of these 90 min is the incubation time, which is a growth delay before the shell starts to grow. The initial shell typically has a thickness of 10–20 nm. The precursor flows to the reactor are stopped, leaving the sample in an H$_2$ atmosphere for 5 min, and this time is used to lower the flows of GeH$_4$ and Si$_2$H$_6$. Growth of the Ge QW follows a similar procedure. Both flows are stopped, and the reactor is flushed for 5 min with H$_2$ to remove residual Si$_2$H$_6$.

For hex-Si$_{0.1}$Ge$_{0.9}$/Si$_{0.3}$Ge$_{0.7}$ QWs, the initial barrier is grown in a single step (Fig. S1d). The remainder of the recipe is comparable to the hex-Ge/Si$_{0.2}$Ge$_{0.8}$ QWs.

## Transmission Electron Microscopy

Transmission Electron Microscopy (TEM) studies were performed using a probe corrected JEOL ARM 200F, operated at 200 kV. All images were acquired at low camera length (8 cm, 68-280 mrad) to minimize the contribution of strain and diffraction contrast.

Energy dispersive X-ray Spectroscopy (EDS) studies were performed using a 100 mm² Centurio EDS silicon drift detector. Quantification of the EDS spectra was done using the Cliff-Lorimer model. The accuracy of EDS quantification was previously confirmed by determining the composition of a single sample, corresponding to MOVPE input $Si_{0.10}Ge_{0.90}$, with both EDS-STEM and Atom Probe Tomography (APT)[18].

Cross-sectional TEM samples of nanowires were prepared using a Focused Ion Beam (FIB) FEI Nova Nanolab 600i Dualbeam system. For this, the NWs were initially swiped from the growth substrate to a piece of Si and then arranged to lie parallel to each other with the aid of a micromanipulator. These NWs were covered with the use of electron-beam induced C and Pt deposition to minimize the ion beam damage in following steps. Afterwards, the NWs were embedded in ion-beam induced Pt deposition. The lamella was cut out by milling with 30 kV Ga ions and thinned down with subsequent steps of 30, 16, and 5 kV ion milling in order to minimize the Ga-induced damage in the regions imaged with TEM.

The QW thickness is mainly determined from images along the [0001] zone axis. QWs of which the thickness could not be measured accurately, due to varying QW position or width within the thickness of the TEM lamella, are excluded from the analysis.

The stacking sequence within the $Ge/Si_{0.2}Ge_{0.8}$ QWs is obtained from Scanning Transmission Electron Microscopy (STEM) images. Within each image, we count the number of planes that have surrounding hexagonal segments. A segment of $i = 1, 2, 3$ planes would represent segments of 2, 3, 4 consecutive neighboring monolayers (ABA,ABAB,ABABA) respectively. Over multiple images, we count how many times we observe a segment that contains $i$ hexagonal stacked planes, which we call $N_i^{Hex}$. Similar reasoning holds for the segments with coherent cubic stacking. The distribution of the hexagonal and cubic segment lengths ($N_i^{Hex}$ and $N_i^{Cub}$ respectively), are shown in Fig. S4a.

The hexagonality $F_i^{Hex}$, i.e., the percentage of the NW that has local hexagonal stacking of at least $i$ planes, as

$$F_i^{Hex} = \frac{\sum_{j=i}^{\infty} j \cdot N_j^{Hex}}{\sum_{j=1}^{\infty} j \cdot N_j^{Hex} + \sum_{j=1}^{\infty} j \cdot N_j^{Cub}} \quad (2)$$

For $i = 1$, above equation calculates the fraction of the NW that is made from hexagonal segments that are *at least* 1 plane long. Longer segments are also included, and weighted according to their length. Higher-order degrees of hexagonality are calculated using larger values of $i$, which are shown in Fig. S4b. The minimum length of a segment with hexagonal stacking, to still have a direct bandgap, is not yet precisely determined.

Local variations of the lattice constant are measured with Geometric Phase Analysis (GPA), utilizing STEM images at atomic resolution. We used a custom, in-house developed toolbox to perform the GPA analysis. The GPA tool calculates the local diffraction pattern, using a 2D-Fourier transformation. Changes of the diffraction spots, due to changes in the local lattice constant, are used to calculate the strain with respect to a reference area. With the 2D-Fourier transformation, it is possible to measure the strain in the horizontal and vertical direction of each image. If the QW is imaged along the [0001] zone axis, this corresponds to the strain in the azimuthal and radial directions of the NW geometry. This reference area is defined within each TEM image, in this case, to be within the inner $Si_{0.2}Ge_{0.8}$ layer.

## X-Ray Diffraction

The X-ray diffraction measurements were made with a Bruker Discover D8. The incidence beam is filtered with a Ge monochromator for the Cu K-$\alpha$ radiation (1.5406 Å). The incidence beam is collimated with a nozzle of 2 mm in diameter. The diffracted beam is measured with a 2D detector, without any optics in between. The 2D detector is used to collect diffracted X-rays with an in-plane angle perpendicular to $2\theta$ of $\pm 0.36°$.

Reciprocal space maps (RSMs) covering the cubic twin [331] until the hexagonal [10$\bar{1}$6] reflection are measured in a single scan. The RSMs are aligned such that the angular coordinates [$\omega$, $2\theta$] of the GaAs [224] substrate reflection correspond exactly to the theoretical values of [61.3474°, 83.7524°].

The hexagonal lattice constants of the NWs are obtained by fitting the RSMs around the [10$\bar{1}$5] reflection with a 2D Gaussian profile. The uncertainty in the peak position of this Gaussian is used to calculate the uncertainty in the lattice constants.

Asymmetrical crystal truncation rods are obtained by taking a line scan along $Q_z$ through the RSMs. The intensity at $Q_x = 1.816$ Å$^{-1}$ is integrated along the $\omega$-direction within a region of $\omega \pm 1.5°$. The range is chosen to collect both the substrate and NW reflections, which occur at slightly different $Q_x$ due to the difference in the in-plane lattice constant.

The asymmetrical crystal truncation rod allows the separation of the hexagonal and the cubic reflections. Hence, it is used as a probe for the amount of hexagonal stacked material within the NWs. One of the main problems with XRD is that it is quite insensitive to the I3 stacking fault, which is the most common defect in the hex-$Si_{1-x}Ge_x$. Consider two hexagonal stacked domains ABAB and BABA, which are aligned along the [0001] axis, separated by either a single "A" plane, i.e., perfect hexagonal stacking, or by a single "C" plane, corresponding to the I3 stacking fault. The only difference between the two configurations is that the I3 defect transforms the local stacking from ABABABABA to ABABCBABA. The two hexagonal domains separated by an I3 defect still interfere constructively, since the I3 defect has no burgers vector[40]. Therefore, we believe that an I3 defect does not broaden any peak in XRD[54]. The I3 stacking, however, should result in a lower intensity of the diffraction signal since there are fewer lattice planes contributing to constructive interference. The relative intensity of the hexagonal peaks between samples is therefore used as a probe for the amount of I3 defects.

To do so, peaks with a Voigt profile are fitted to the asymmetrical crystal truncation rods. Near the hex-[10$\bar{1}$5] peak, two peaks are fitted. One around $Q_z \approx 4.82$ Å$^{-1}$, which we attribute to signal coming from the core-shell NWs, and one around $Q_z \approx 4.78$ Å$^{-1}$, which we attribute to bulk-like WZ GaAs, that parasitically grows on the GaAs substrate around the base of the NW. After $Si_{1-x}Ge_x$ shell growth, this bulk-like WZ GaAs maintains a lattice constant close to WZ GaAs, while the lattice constant from the NW is shifted towards $Si_{1-x}Ge_x$. The obtained hex-[10$\bar{1}$5] peak areas are normalized to the [224] substrate reflection, to account for small imperfections in the alignment between the samples. Moreover, the [10$\bar{1}$5] peak areas are divided by the volume of the NWs. These volumes are calculated from the length and diameter, as extracted from SEM images. When normalized in this manner, all GaAs-$Si_{1-x}Ge_x$ core-shell NW samples give a similar number within a factor of 1.5 (Fig. S4d).

## Photoluminescence

The (macro) PL measurements were performed using a Thermo Scientific iS50R step-scan Fourier Transform InfraRed Spectrometer (FTIR). The as-grown NW samples are introduced to the setup by placing them in a LHe cooled Oxford Instruments HiRes2 continuous-flow cryostat which can be temperature controlled using the integrated heater governed by an Oxford Instruments MercuryiTC. The samples are excited using a Quasi-continuous wave (Quasi-CW) 976 nm laser,

focused on the sample by a 2.1 cm focal distance off-axis parabolic Au mirror to an ≈ 100 μm spot and the collected photoluminescence is measured using the internal Mercury Cadmium Telluride (MCT) detector of the FTIR. The excitation laser was filtered out using a germanium window (1950 nm) or a 1650 nm long pass filter. To extract the NW response from the black-body radiation background, the laser is modulated using a 38 kHz square wave generated by a Siglent SDG1032X Arbitrary Waveform Generator (AWG) and the signal is finally demodulated using a Zurich Instruments MFLI Lock-in Amplifier (LIA). To improve the stability of the modulation frequency, the AWG was locked to the oscillator in the LIA using the 10 MHz clock signal reference.

For Fig. 3, the QW and reference samples were measured at the lowest excitation density that still gave an acceptable Signal-to-noise ratio, being 3, 13, 50, 39, 6, 9 and 13 W cm$^{-2}$ for the 9, 6, 4, 3, 2.5, 2 and 1.5 min QWs and 64 and 2 W cm$^{-2}$ for the bulk hex-Si$_{0.2}$Ge$_{0.8}$ and hex-Ge reference samples respectively and lightly smoothed for clarity using a 21 point, linear Savitzky-Golay filter. The finite quantum well model added to Fig. 3b was calculated using the bulk effective masses for the well and interpolated effective masses between bulk hex-Ge ($m_e \approx 0.079\, m_0$, $m_h \approx 0.055\, m_0$)[19] and hex-Si for the barrier ($m_e = 0.122\, m_0$, $m_h = 0.213\, m_0$)[50]. The bandgap energy $E_{Well}$ was determined from the experimental 0.354 eV emission peak of the hex-Ge reference spectrum increased by 13 meV to account for the shift due to strain from the QP calculations and $E_{Barrier} = 0.570$ eV was determined from the peak energy of the hex-Si$_{0.2}$Ge$_{0.8}$ reference spectrum, the reference spectra are shown in Fig. 3a. The band-offsets were assumed to be symmetrical as indicated by the QP calculations Fig. S10c and the experimental estimation Fig. S6d.

For Fig. 5, the QW and reference samples were measured at 0.88 kW cm$^{-2}$ for the 5 and 15 min QWs and 0.88 and 0.42 kW cm$^{-2}$ for the bulk Si$_{0.1}$Ge$_{0.9}$ and Si$_{0.3}$Ge$_{0.7}$ reference samples respectively. The spectra were background corrected by fitting the sum of an exponential Urbach tail[55,56] from the GaAs epitaxial substrate and a Gaussian peak spectrum for each spectrum, after which the exponential is subtracted. As the Si$_{0.3}$Ge$_{0.7}$ reference had a very low intensity even at high excitation density it was smoothed for clarity after the baseline correction using an 81 point, quadratic Savitzky-Golay filter. The spectra of the 5 min QW and Si$_{0.3}$Ge$_{0.7}$ reference after baseline correction are in agreement with the μPL spectra of single NWs mechanically transferred onto an Aluminum-Nitride (AlN) substrate shown in Fig. S9. The finite quantum well model added to Fig. 5c was calculated using interpolated bulk effective masses between hex-Ge ($m_e = 0.076\, m_0$, $m_h = 0.055\, m_0$)[19] and hex-Si ($m_e = 0.122\, m_0$, $m_h = 0.213\, m_0$) for both the barrier and the well material, the band-offsets were assumed to be symmetric and determined from the experimental emission energies of the well and barrier reference samples shown in Fig. 5b.

## Time-resolved photoluminescence

The single nanowire spectrum is investigated using a Time-Resolved Fourier-Transform-Infrared-Spectroscopy setup (TR-FTIR). This setup allows us to study the spectrally-resolved time decay of the photoluminescence of a sample. The as-grown hex-SiGe NWs samples are mechanically transferred on a planar AlN substrate and are introduced to the setup by placing them in a LHe cooled Oxford Instruments HiRes2 continuous-flow cryostat. The temperature is set to 4 K using an Oxford Instruments MercuryiTC. The samples are optically excited using a femto-second pulsed mode-locked fiber laser (NKT ORIGAMI 10−40) with a wavelength of 1032 nm and repetition rate of 40 MHz. A 36x/0.40NA Cassegrain objective is used to excite and collect the signal from the sample. The excitation/collection spot diameter on the sample is 3 μm. The PL signal from the sample is sent through the Nireos GEMINI birefringent Fourier transform interferometer to acquire spectrally resolved photoluminescence and finally collected by a Superconducting Nanowire

Single-Photon Detector (SNSPD) with a measurement window up to 2.35 μm (Single Quantum EOS110). A 1350 nm long-pass filter is placed before the GEMINI module to block the excitation laser reflected on the sample. For the single NW lifetime measurement, the GEMINI interferometer is kept fixed at the zero path distance and the measurement is performed without acquiring spectral information from the NW signal.

## Theoretical and numerical methods

All calculations were performed within the framework of Density Functional Theory (DFT) using the VASP software[57,58] and the projector-augmented wave method[59], with a plane-wave cutoff of 500 eV. The shallow 3d levels of Ge were treated as valence states. Geometry relaxations employed the Perdew-Becke-Ernzerhof exchange-correlation (XC) functional PBEsol[60]. Brillouin zone integrations were carried out with a Γ-centered 12 × 12 × 6 k-point grid for lonsdaleite (2H) crystals. Quasiparticle band structures were computed using the MBJLDA XC potential of Tran and Blaha[61], which combines the modified Becke-Johnson (MBJ) exchange[62] with correlation in the local density approximation (LDA)[63]. Spin-orbit coupling (SOC) was consistently considered, as the resulting corrections to the band structure are crucial for Ge and alloys with a substantial Ge content. Branch point energies were calculated following the method of reference[46], and they were applied whenever necessary to align energy levels of different materials and heterostructures. This approach was already validated for [0001] interfaces in reference[47]. The resulting band structures of hex-Ge and hex-SiGe alloys are consistent with previously published results[19,20,64]. Numerical differences between the reported findings here and those published earlier stem from the additional biaxial strain applied in this work to replicate experimental conditions, as discussed in the main text.

In our approach, the Ge layer thickness affects the lowest conduction band of the bulk Si$_{0.25}$Ge$_{0.75}$ barrier material (red solid line in Fig. 6d). The structural optimization within the DFT approach of the studied MQW structures gives rise to mutual biaxial strains in the hex-Ge well layers as well as in the SiGe barrier layers in dependence of the layer thicknesses in addition to the significant "external" biaxial strain taken from the measurements. Despite this strong biaxial strain due to the assumed pseudomorphic growth of the hex-Ge/SiGe heterosystems on the wurtzite-GaAs core wires, the additional small strain distribution in the heterosystem only slightly affects the actual strong strain situation in the barrier material resulting in small band edge variations made visible in Fig. 6d by a red (blue) line for electrons (holes). The accompanying changes of the QW barrier heights of less than 0.015 eV hardly influence the carrier confinement in the lowest $n = 1$ levels in the QWs.

## Reporting summary

Further information on research design is available in the Nature Portfolio Reporting Summary linked to this article.

## Data availability

The raw data generated in this study have been deposited in Zenodo: https://doi.org/10.5281/zenodo.10839570.

## Code availability

The VASP code used for electronic structure calculations can be acquired from the VASP Software GmbH at https://www.vasp.at/. The Python code used for the analysis of the growth and the photoluminesence experiments is provided as 'Source Code' files deposited in Zenodo: https://doi.org/10.5281/zenodo.10839570.

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

## Acknowledgements

We thank P.J. van Veldhoven and M.G. van Dijstelbloem for the technical support of the MOVPE reactor. We thank Orson A.H. van der Molen for the GPA analysis. This project received funding from the European Union's Horizon 2020 research and innovation program under grant agreement number 964191, Opto Silicon (W.H.J.P., M.M.J., S.B., F.B., J.E.M.H., and E.P.A.M.B.), the Dutch Organization for Scientific Research (NWO) in the Zwaartekracht Project, Grant No. 024.002.033 (M.A.J.T.), 739.017.002, (V.T.L.), and OCENW.M.21.052 (R.F.) and Solliance and the Dutch province of Noord-Brabant for funding the TEM facility.

## Author contributions

W.H.J.P., M.M.J. carried out the growth of hex-$Si_{1-x}Ge_x$/$Si_{1-y}Ge_y$ quantum wells. W.H.J.P. analyzed the data. V.T.L. and M.C.H. carried out the photoluminescence spectroscopy and analyzed the optical data. R.F. and M.A.J.T. performed time-resolved spectroscopy on single quantum well nanowires. A.B. and F.B. performed the DFT calculations. W.H.J.P. performed the XRD measurements. W.H.J.P. and M.M.J. performed the FIB cuts, and M.A.V. performed the TEM analysis. S.B., F.B., J.E.M.H., and E.P.A.M.B. supervised the project. F.B. contributed to the interpretation of data, and W.H.J.P., V.T.L., F.B., J.E.M.H., and E.P.A.M.B. contributed to the writing of the manuscript. All authors discussed the results and commented on the manuscript.

## Competing interests

The authors declare no competing interests.
