## [Peer Review File · Nature Communications]

Direct bandgap quantum wells in hexagonal Silicon GermaniumREVIEWER COMMENTS

Reviewer #1 (Remarks to the Author):

This paper presents the results of photoluminescence experiments on quantum wells realized using the hexagonal SiGe shell of GaAs/SiGe core/shell nanowires. The realization of hexagonal SiGe heterostructures is a necessary step towards the application of the hexagonal diamond phase of SiGe for optoelectronic devices. Thus the work is certainly of significance to the field and contains several elements of novelty, being suitable for publication in Nature Communications.

However, there are some points to be addressed by the authors, and many modifications to the paper are necessary before publication.

1. It is not clear to me why the paper started presenting the theoretical results. Even in the abstract, the authors wrote that the ab initio bandstructure calculations support the experimental findings. In fact, the strain values imposed experimentally are derived from the experimental measurements, however these are presented later. Also, the calculations of the heterostructures and the band alignment illustrated in Fig. 1 would be easier to understand if the sketch of the experimental nanowires and the other experimental results illustrated in Fig. 2 were presented before the theory part.

2. Continuing with the biaxial strain used in the simulations of the Ge/SiGe heterostructures: what about the perpendicular direction to the interface? Is the deformation along the [1-100] direction considered, or the unstrained lattice parameter along this direction has been used? Finally, I can't understand how these strain values are derived from experiments (from Fig E6?).

3. One should clearly explain that and why the composition of the SiGe in the theoretical models is different than the experimental ones. Starting with a Si_{0.25}Ge_{0.75} model and then reading a Ge/Si_{0.2}Ge_{0.8} experimental heterostructure is a bit confusing for the reader.

4. Page 2, lines 114-118: the authors list some values of the confinement energy for electrons and holes, but it is not explained anywhere how it is calculated. Then, because most of the hexagonal segments in the [0001] direction are thinner than 4 nm (shown in Figure E5a), should one take into account also the quantum confinement in this direction due to cubic/hexagonal heterostructures?

5. Page 2, lines 138-140: "As illustrated, the thickness of the Ge QW varies between the different facets". This is not very clear in Fig E3a, why not use different colours for the different facets?

6. Page 2-3, lines 160-164: The authors wrote that a broad distribution in the length of segments with the hexagonal stacking is observed, in contrast to the cubic ones. This is particularly true in Fig E5a, but using a log scale. Actually, plotting an integrated density of the segment lengths one should see that the majority of both hexagonal and cubic segments are composed of only 2, 3 or 4 monolayers. This opens an important question on the quality of these nanowires that should be discussed by the authors. Could one discuss the hexagonal phase only, but we do see very many alternating hexagonal-like and cubic-like layers in the shell? Could this be neglected when considering the quantum wells in the radial direction?

7. Page 3, lines 170-180: This paragraph on the pseudomorphic strain is not easy to understand, especially because some fundamental images are only as extended data and with no discussion.

8. Page 4: The reason for the two peaks observed in the PL spectrum of the thicker QW should be further discussed. It is not yet very convincing that it is due to the presence of distinct QW thickness at different facets, and Fig E3b do not clearly show this.

9. One of the most critical points of the paper is that there are too many figures that are often fundamental to understanding the discussion, but provided as extended data figures. Often the caption with these images is not enough and further discussion would be useful. In general, the paper and the extended data figures should be more homogenous. For Instance, why in Fig 4a the PL spectrum is that of a 10 nm NW (2.5 min) but the extended analysis of the thickness in Fig. E3 and E4 are on different NWs? Fig. 2 seems also to focus on a different NW, while Fig. 3 is not clear.

Reviewer #2 (Remarks to the Author):

The work deals with group-IV semiconductors in the hexagonal crystal phase, a topic that sparked high interest recently. In this work, the authors go a step further down in dimensionality compared to their previous work by demonstrating that quantum wells can indeed be embedded into the hexagonal SiGe shell of the grown wires. Thus, this work provides a pivotal first step in adding functionality to these systems of hexagonal group-IV semiconductors. The experimental methodology is thorough and, mixed with the theoretical insights, convincing.

There are, however, some points that should be clarified and addressed by the authors.

1) Throughout the manuscript (ms), it is claimed that the system has a direct band gap - based on temperature-dependent PL intensity decay (p4), and, e.g., the 1 ns carrier lifetime (p5), comparable to the radiative (sic!) lifetime of hex-SiGe bulk.

That is a statement to be careful with. In TCSPC, the fastest recombination process always dominates the spectral shape. And it can be radiative and non-radiative. The authors cannot exclude non-radiative transitions (presence of I3 defects, interfaces, twins, segmented heterostructures). A correlation of the power-dependent PL measurements (Fig. E7(b)) with the TCSPC measurements suggests that non-radiative decay cannot be neglected. The slopes of the power-dep. measurements are clearly sub-linear, meaning the carriers must be lost non-radiatively, even at low excitation densities (Fig. E7(b)). Therefore, also the single exponential decay in the TCSPC measurements (fig. E10(b)) can be indicating in reality the non-radiative lifetime.

It is not to be said that the hex-Ge QWs are not direct, but the experiments are not providing sufficient proof that they are indeed.

2) Intermixing was not taken into account for the QWs. For the used high growth temperatures of $\sim 600^\circ\text{C}$, significant Ge surface segregation has to be expected (which might be seen in Fig. 2(c) in the form of very blurry interfaces. This would change the QW shape and shift the emission to higher energies as compared to the theoretical results (Fig. 3(b)).

3) Doping was not taken into account. This is somewhat surprising since the authors have stated in their earlier Nature paper (Fadaly et al.) that the presence of the GaAs core dopes the hex-group-IV to $\sim 10^{19} \text{cm}^{-3}$. This will likely influence the optoelectronic behavior of the transitions. Please indicate why this can be neglected.

Minor 4) The mix of providing the layer thickness in nm and minutes is confusing and disturbs the reading flow. Please correct to nm throughout.

Minor 5) In the abstract, the authors state that Si is a poor choice for active photonic applications. Modulators and Detectors are active and commercially available. Only emission is a problem. The authors might want to change that.

Reviewer #3 (Remarks to the Author):

Referee report

on the paper of W.H.J. Peeters et. al. "Direct bandgap quantum wells in hexagonal Silicon Germanium"

The current manuscript represents the development of the previously published idea (see, for example, Nature 580, 205 (2020)) devoted to the Ge and SiGe band structure modification via creating the hexagonal form of such materials. According to the previous results, Ge and SiGe in the hexagonal arrangement should be the direct-gap semiconductors and this could help to overcome the main problem of their inherently low light emission efficiency and so potentially enable the construction of Si-compatible light source using group-IV materials. In particular, in the current manuscript it is proposed to use somewhat "classical" approach of double heterostructure - the Ge QW (or SiGe QW with low Si fraction) as an active medium that is sandwiched between SiGe barrier layers.

In the current manuscript thorough theoretical investigation of bandgap modification of hexagonal Ge/SiGe structures was presented and rather large amount of experimental data was obtained. Analysis of the hex-Ge/SiGe photoluminescence (PL), in particular, dependencies of the PL peak position on the hex-Ge thickness and on the excitation density has proven the realization of type-I heterojunction in which both electrons and holes are confined in the hex-Ge QW region. The latter is of indisputable importance.

However, I have significant objections related to the goal setting of this work, as it was stated that it is aiming "for integrating light emitting materials on silicon". Indeed, all studied structures were fabricated on GaAs(111)B substrates, but not on silicon, and so usage of such substrate type and orientation would allow only the heterogeneous integration with Si or SOI platforms. As such, the fabrication procedure of hex-Ge/SiGe NW-based structures, presented in the manuscript, does not seem any easier (if it's heavier) than integration of "conventional" planar A3B5 structures on silicon. To the contrary, both heterogeneous and hybrid or even monolithic integration of A3B5 QW-and QD-based light-emitting structures on silicon was very thoroughly studied and now they are already used in real optoelectronics devices. It seems questionable that the GaAs NW-based hex-Ge structures would be easier and cheaper (!) to integrate with silicon as compared to the former "conventional" A3B5-based ones.

Moreover, as regards to group-IV materials, it seems that usage of strained-Ge or GeSn structures which could be directly fabricated on silicon would be more promising approach as compared to hex-Ge NW-based ones. Both strained-Ge and GeSn systems has already shown the laser action at cryogenic and even at room temperature (see, for example, Nat.

Commun. 10:2724 (2019); Nat. Phot. 14, 375 (2020); Optica 7, 924 (2020); APL 120, 051107 (2022)). Electrical pumping which is needed in real devices was also demonstrated. Furthermore, I have not found any significant advances in light-emitting properties of hex-Ge/SiGe structures, reported in this manuscript, compared to the previous work published in Nature 580, 205 (2020).

I have also several smaller remarks:

1. Some difference between the hex-Ge/SiGe structures used for theoretical calculations and for experimental examination exist. Band structure was calculated for MQW hex-Ge/SiGe structures having thin SiGe barriers and thicker Ge QWs, while single-Ge QW structures embedded into thicker SiGe barriers were experimentally fabricated and studied. This could potentially mislead the readers.
 2. It is not clear from theoretical calculations why the Ge layer thickness affects the lowest conduction band of the bulk $\text{Si}_{0.25}\text{Ge}_{0.75}$ barrier material (red solid line in Fig. 1d). It is also unclear what is the reason for the different behavior of the calculated values for electron and hole band offsets of Ge/Si $_{0.25}$ Ge $_{0.75}$ structures (Fig. E2c in extended data).
 3. From the reported results it seems that lattice constants of NW hex-Ge/Si $_{0.2}$ Ge $_{0.8}$ are close to lattice constants of bulk hex-Si $_{0.2}$ Ge $_{0.8}$, and strain relaxation in Ge layer does not generate defects (Fig. E6 in extended data). But how the strain relaxation between GaAs NW and hex-Si $_{0.2}$ Ge $_{0.8}$ shell occurs?
 4. Large spread of hex-Ge QW thickness in a single NW and cubic phase inclusions make the PL interpretation questionable. According to the Fig. 3b in main text and Fig. E3b in extended data, the Ge layer after 9 min of growth possesses larger thickness deviation and smaller average thickness, than the similar Ge layer after 6 min of growth. At the same time, the PL peak related to the structure with 9 min of Ge growth has smaller width and is blueshifted as compared to PL peak related to the structure with 6 min Ge growth. I think, there is a contradiction here.
 5. It seems that low temperature PL spectra presented for 9 min Ge growth structure in Fig.4c possess the pronounced double-hump view, while the PL spectrum for the same structure presented in Fig.3a (red line) is much more like a single-hump one. Please comment.
- Also, the PL temperature quenching for different structures should be discussed in more detail.
6. The studied structures have various defects (cubic phase inclusions, twins etc) as reported by authors. So it seems questionable that the measured PL decay time is determined precisely by the radiative, and not the non-radiative processes.

Based on the considerations summarized above I recommend to send this article to some other journal.

Dear Reviewers,

We thank all the reviewers for their comments which helped us to significantly improve the manuscript. The most important changes are summarized below.

- Following the suggestion of reviewers 1 and 3, we shifted the theory part to the end of the paper. The old Fig. 1 is now Fig. 6.
- We have added a new Fig. 1 to introduce our coaxial nanowire quantum wells to the reader.
- In response to the reviewers, we have added several new figure panels. These new figure panels are Fig. 1b, Fig. 4 b, 4d. We also modified Fig. 4f.
- As suggested by reviewer 1, we have moved two figure panels (Figs. 2b and 4g) from the extended data to the main text to improve the readability. Figure 4 is now systematically showing all data from the 10 nm and the 24 nm samples, while the Extended Data shows the temperature dependence and the light-in-light-out curves for all QW thicknesses.
- We added a Discussion section.
- We significantly expanded the Conclusions section.
- In response to reviewer 2, we added extended evidence for direct bandgap emission in the quantum wells by expanding the analysis of the measured carrier recombination lifetime.
- Following the suggestion of reviewer 3, we analyzed the thermal quenching and deduced an independent experimental assessment of the value of the conduction band offset.

In addition to highlighting these major changes, we answer the reviewers point-by-point below. We emphasize that these remarks also generated many small improvements of the manuscript.

Reviewer #1 (Remarks to the Author):

This paper presents the results of photoluminescence experiments on quantum wells realized using the hexagonal SiGe shell of GaAs/SiGe core/shell nanowires. The realization of hexagonal SiGe heterostructures is a necessary step towards the application of the hexagonal diamond phase of SiGe for optoelectronic devices. The work is certainly of significance to the field and contains several elements of novelty, being suitable for publication in Nature Communications. However, there are some points to be addressed by the authors, and many modifications to the paper are necessary before publication.

1. It is not clear to me why the paper started presenting the theoretical results. Even in the abstract, the authors wrote that the ab initio bandstructure calculations support the experimental findings. In fact, the strain values imposed experimentally are derived from the experimental measurements, however these are presented later. Also, the calculations of the heterostructures and the band alignment illustrated in Fig. 1 (Fig. 6 in the revised version) would be easier to understand if the sketch of the experimental nanowires and the other experimental results illustrated in Fig. 2 were presented before the theory part.

We like to thank the reviewer for this comment. Indeed, the theoretical calculations used some of the experimental parameters as input. Therefore, the flow of the text has been changed, such that the experimental part is now discussed before the band structure calculations, which have been shifted towards the end of the paper. The suggestion of the reviewer improved the readability of the manuscript.

2. Continuing with the biaxial strain used in the simulations of the Ge/SiGe heterostructures: what about the perpendicular direction to the interface? Is the deformation along the [1-100] direction considered, or the unstrained lattice parameter along this direction has been used? Finally, I can't understand how these strain values are derived from experiments (from Fig E6? (Now E5)).

For the theoretical calculations, we assume pseudomorphic growth of the investigated Ge/SiGe heterostructures on the (1-100) facets of the wurtzite-GaAs core nanowires, which is consistent with experiments. The a and c lattice constants of Ge are close to those of GaAs, while those of $\text{Si}_{0.25}\text{Ge}_{0.75}$ are approximately 1% smaller compared to GaAs. Consequently, the hex-Ge and $\text{Si}_{0.25}\text{Ge}_{0.75}$ layers forming the QW structures and oriented in [1-100] directions (see Fig. 6a for the unit cell with the crystal directions) are biaxially strained compared to their bulk situations. The rectangular 2D primitive unit cells have edges along the [11-20] and [0001] directions with edge lengths corresponding to the bulk a and c lattice constants, respectively. The lattice constants of the experimentally studied Ge QW layers are compressed (see Fig. 2d) as described in the text. Indeed, the accommodation of the edge lengths of the rectangular unit cells of the core and shell materials gives rise to lattice constants around $a=3.96$ and $c=6.517$ (see Extended data Fig E5a), which correspond to a compressive strain of -0.6 % and -0.91% relative to unstrained hex-Ge. According to the assumption of pseudomorphic growth the atomic positions in [1-100] growth direction follow according to the inverse Poisson effect, as observed in Extended data E5d. This also happens in the DFT calculations, where the atomic positions in the plane perpendicular to the [1-100] direction are fixed according to the measurements but the coordinates parallel to [1-100] are allowed to relax freely.

The experimental observations are now described in more detail in lines 151-158.

3. One should clearly explain that and why the composition of the SiGe in the theoretical models is different than the experimental ones. Starting with a $\text{Si}_{0.25}\text{Ge}_{0.75}$ model and then reading a Ge/ $\text{Si}_{0.2}\text{Ge}_{0.8}$ experimental heterostructure is a bit confusing for the reader.

We thank the reviewer for this comment. Since the 3D primitive unit cell of lonsdaleite Ge contains four atoms, the $\text{Si}_{0.2}\text{Ge}_{0.8}$ alloy forming the barriers of a Ge well system can be most closely approximated by an ordered arrangement of the Si and Ge atoms with atom ratio $\frac{1}{4}=0.25$. Besides the linear arrangement of the Si atoms in [1-100] direction as illustrated in Fig. 6a we have studied also other ordered arrangements of Si atoms, e.g. with zigzag chains and Si dimers, which however give only a negligible variation of the confinement energies by 5 meV or less. The increase of the Si composition x in the barrier materials by 0.05 only slightly increases the barrier heights for electrons and holes. This increase will practically not influence the energies of the lowest $n=1$ levels in the QW structures.

The reasons and the details of the modeling of the SiGe barrier materials within the ab initio calculations are described in the text of the resubmitted manuscript, starting at line 339 by "The Ge/ $\text{Si}_{0.25}\text{Ge}_{0.75}$ system is the closest approximation of the experimentally realized Ge/ $\text{Si}_{0.2}\text{Ge}_{0.8}$ QWs, which still allows modeling of the alloy barriers by ordered arrangements of a single Si and three Ge atoms in one Lonsdaleite unit cell."

4. Page 2, lines 114-118: the authors list some values of the confinement energy for electrons and holes, but it is not explained anywhere how it is calculated. Then, because most of the hexagonal segments in the [0001] direction are thinner than 4 nm (shown in Figure E5a {Figure E4a in the

revised version)), should one take into account also the quantum confinement in this direction due to cubic/hexagonal heterostructures?

The confinement energies of electrons and holes and their extraction from the band structures of the MQW systems and the strained hex-Ge bulk are now described in the text of the resubmitted manuscript, at line 391-405. Moreover, the derivation of the effective masses in the quantization direction using an effective mass approximation and a quantum well model are now described in more detail, starting from line 406.

We agree with the observation of the reviewer concerning the bilayer stacking in [0001] direction, which may vary between AB lonsdaleite (2H) and ABC diamond (3C) structure or even other hexagonal polytypes 4H, 6H, ... However, these effects happen perpendicular to the growth and quantization direction [1-100]. The stacking fluctuations in [0001] direction do not influence the confinement of electrons and holes in QW [1-100] direction due to the barrier formation by Si incorporation. According to A. Belabbes et al. in PRB 106, 085303 (2022) and PRB 107, 039903 (2023) the stacking fluctuations in [0001] direction may tend to the formation of one-dimensional quantum wires with an additional confinement in perpendicular [0001] direction due to band offsets between 3C and 2H, instead of the 2D electronic systems of the QWs. The additional confinement in [0001] direction may partly explain the slightly larger QW emission energies found experimentally compared to the theoretical predictions in Fig. 3b, obtained without additional confinement in the perpendicular direction.

In response to the reviewer, the possibility of additional confinement due to stacking fluctuations along the [0001] direction is now included in the discussion section of the main text, starting from line 443.

5. Page 2, lines 138-140: "As illustrated, the thickness of the Ge QW varies between the different facets". This is not very clear in Fig E3a (Now E2a), why not use different colours for the different facets?

Since we study coaxial nanowire QWs, we identify two types of fluctuations in the QW thickness. First, there are intra-wire fluctuations: the QW thickness varies within the same NW, since growth takes place on 6 facets. Secondly, there are wire-to-wire fluctuations. We noted that the variation on the same nanowire is much larger than the wire-to-wire fluctuations. An example is illustrated in Fig 2a. Total statistics of all measured QW thicknesses are included in 2b, which is a figure that has been shifted from the extended data to the main text.

Determining the type of fluctuations is important. The QWs on the different facets of the same NW are all connected, meaning that carriers might diffuse towards the largest QW thickness.

An additional line has been added around line 110-114 to highlight which type of QW fluctuations is largest in these samples: "The facet-to-facet fluctuation within one NW dominates the distribution of QW thicknesses, it is larger than the deviation in average QW thickness between different NWs of the same sample"

6. Page 2-3, lines 160-164: The authors wrote that a broad distribution in the length of segments with the hexagonal stacking is observed, in contrast to the cubic ones. This is particularly true in Fig E5a (Figure E4a in the revised version), but using a log scale. Actually, plotting an integrated density of the segment lengths one should see that the majority of both hexagonal and cubic

segments are composed of only 2, 3 or 4 monolayers. This opens an important question on the quality of these nanowires that should be discussed by the authors. Could one discuss the hexagonal phase only, but we do see very many alternating hexagonal-like and cubic-like layers in the shell? Could this be neglected when considering the quantum wells in the radial direction?

We first emphasize that this manuscript is intended to be a proof-of-concept paper showing the possibility to grow direct bandgap type I QWs. The observed shift of PL emission with QW thickness (Fig. 3 of the main text), clearly indicates the existence of radial carrier confinement. A further optimization of the material quality should be the subject of a follow-up paper.

It is an important point raised by the reviewer in how many directions the carriers are confined, only in the radial, or also in the longitudinal direction of the nanowire? Showing an integrated density of the segments is a good suggestion from the reviewer. An additional panel has been added in Fig E4b, which shows this. It is not correct to state that the majority of the nanowire consists of 2,3 or 4 monolayer segments of hexagonal stacking, since the longer segments have to be weighted higher. For example, 30% of the NW consists of hexagonal segments that are at least 9 monolayers long, equivalent to 3nm.

We note that hex-cubic homojunction of Ge is expected to be of type I. Thus, cubic sections would act as barriers for both holes and electrons. However, as noted in Fig E4a, these cubic segments are typically only 1 nm wide. These thin segments thus do not act as barriers, but they tend to slightly increase the emission energy. We now discuss this point in the Discussion section line 440-450 in the modified manuscript.

When comparing different samples, we expect that each QW sample has a similar distribution of hexagonal-like and cubic-like layers, based on the XRD measurements presented in Fig E4c-d. Thus, additional confinement energy in the longitudinal direction due to cubic-like stacking should be equal for all samples, which would be consistent with the additional confinement observed when comparing experiment and theory. We addressed this point explicitly in the main text around line 220, as well as in the Discussion section between lines 443-452.

7. Page 3, lines 170-180: This paragraph on the pseudomorphic strain is not easy to understand, especially because some fundamental images are only as extended data and with no discussion.

We agree with the reviewer that this paragraph indeed was not easy to understand. We decided to keep the images in the extended data, but to expand the discussion in the manuscript. The method of determining the local variations from TEM images with Geometric Phase analysis has been included in the methods section. A schematic has been added in E6 to highlight the radial/azimuthal direction as shown in the experimental panels.

The main text has been adjusted from, :

“Increasing the Ge thickness does not significantly influence the lattice parameters of the NWs (Extended data Fig. E5a), so all studied samples have comparable c-lattice constants in the Ge QW. However, the Ge is compressed along the $\langle 1120 \rangle$ and $\langle 0001 \rangle$ directions, and pseudomorphic strain relaxation in the Ge QW results in an increased radial relaxation with increasing thickness, as confirmed by the Geometric Phase Analysis (GPA) of TEM images (Extended data Fig. E5b).”

To (see lines 144-150)

“Increasing the Ge thickness does not significantly influence the lattice parameters of the NWs (Extended data Fig. E5a), so all studied samples have comparable c-lattice constants in the Ge QW. The $\text{Si}_{0.2}\text{Ge}_{0.8}$ barriers have smaller lattice constants than the Ge in the QW, and the Ge is therefore compressed along the $\langle 1120 \rangle$ and $\langle 0001 \rangle$ directions. Pseudomorphic strain relaxation in the Ge QW results in an increased lattice constant along the $\langle 1100 \rangle$ direction. This radial relaxation becomes more pronounced if the Ge thickness is increased, as confirmed by the Geometric Phase Analysis (GPA) of TEM images (Extended data Fig. E5b)

8. Page 4: The reason for the two peaks observed in the PL spectrum of the thicker QW should be further discussed. It is not yet very convincing that it is due to the presence of distinct QW thickness at different facets, and Fig E3b (Now Fig. 2b) do not clearly show this.

In the 24 nm quantum well sample a broad distribution of the QW thicknesses is found, which is now shown in the main text as Fig. 2b. This broad distribution of QW thicknesses is one of the possible explanations for a double peak spectrum at high excitation power. It is however also possible that the double peak spectrum is mainly due to the observation of the HH2-CB2 transition in the wide QW. The exact origin of the double peak observed at high excitation densities is currently hard to prove and beyond the scope of the present paper. We added this point more clearly to the main text in lines 212-221.

9. One of the most critical points of the paper is that there are too many figures that are often fundamental to understanding the discussion, but provided as extended data figures. Often the caption with these images is not enough and further discussion would be useful. In general, the paper and the extended data figures should be more homogenous. For Instance, why in Fig 4a the PL spectrum is that of a 10 nm NW (2.5 min) but the extended analysis of the thickness in Fig. E3 and E4 (Now Fig. E2 and E3) are on different NWs? Fig. 2 seems also to focus on a different NW, while Fig. 3 is not clear.

We truly thank the reviewer for this remark! The new Fig. 4 shows both the excitation power dependence of the narrow and the wide QW in panels 4a,d and the temperature dependence in panels 4b,e. Moreover, we moved data from the Extended data towards the main text to show the Arrhenius plots for both samples in Fig. 4f and the light-in-light-out plots for both samples in Fig. 4g. The QW thicknesses have also been moved from the Extended Data to Fig. 2b. This makes Fig. 4 much more homogeneous.

We emphasize that both Fig. 2b and Fig. 3 show all hex-Ge/SiGe samples discussed in this paper, while Fig. 4 is focused to the 10 nm thin QW and the 24 nm wide QW. Figs. E2 and E3 show TEM images taken from the same nanowire samples as displayed in Fig. 3 of the main text.

Reviewer #2 (Remarks to the Author):

The work deals with group-IV semiconductors in the hexagonal crystal phase, a topic that sparked high interest recently. In this work, the authors go a step further down in dimensionality compared to their previous work by demonstrating that quantum wells can indeed be embedded into the hexagonal SiGe shell of the grown wires. Thus, this work provides a pivotal first step in adding functionality to these systems of hexagonal group-IV semiconductors. The experimental

methodology is thorough and, mixed with the theoretical insights, convincing. There are, however, some points that should be clarified and addressed by the authors.

1) Throughout the manuscript (ms), it is claimed that the system has a direct band gap - based on temperature-dependent PL intensity decay (p4), and, e.g., the 1 ns carrier lifetime (p5), comparable to the radiative (sic!) lifetime of hex-SiGe bulk.

That is a statement to be careful with. In TCSPC, the fastest recombination process always dominates the spectral shape. And it can be radiative and non-radiative. The authors cannot exclude non-radiative transitions (presence of I₃ defects, interfaces, twins, segmented heterostructures). A correlation of the power-dependent PL measurements (Fig. E7(b) (Fig. E6b in the resubmitted manuscript) with the TCSPC measurements suggests that non-radiative decay cannot be neglected. The slopes of the power-dep. Measurements are clearly sub-linear, meaning the carriers must be lost non-radiatively, even at low excitation densities (Fig. E7(b) (Now Fig. E6b)). Therefore, also the single exponential decay in the TCSPC measurements (fig. E10(b) (Now Fig. E9b) can be indicating in reality the non-radiative lifetime.

It is not to be said that the hex-Ge QWs are not direct, but the experiments are not providing sufficient proof that they are indeed.

We thank the reviewer for this comment, which is indeed very important since theoretical DFT calculations show a radiative lifetime of 20 μs for hex-Ge, which comprises the well material of our hex-Ge/SiGe QWs. This comment clearly shows that we did not provide sufficient proof for a direct bandgap in the initial manuscript.

Semiconductors generally show almost pure radiative recombination at low enough temperature. The PL-intensity plotted in Figs. 4f and E6c, is proportional to the internal quantum efficiency for radiative emission, which varies with temperature according to the ratio of the radiative recombination rate, divided by the total recombination rate $\eta_{int} = \tau_r^{-1} / (\tau_r^{-1} + \tau_{nr}^{-1}(T))$, where the nonradiative recombination rate is thermally activated by $\tau_{nr}^{-1}(T) = \tau_{nr,0}^{-1} e^{-E_a/kT}$ similar to III-V materials⁵⁻⁹. At low enough temperature, the nonradiative recombination rate will vanish and the PL-efficiency will not further increase with decreasing temperature as shown in Fig. E6c. This implies that, for our hex-Ge/SiGe QW, the radiative limit is reached below 10K at an CW excitation density of $P = 0.88 \text{ kW/cm}^2$. Our time-resolved PL measurements employ an excitation density between 0.027 and 0.442 mJ/cm², which corresponds to a CW excitation density of roughly $P = 27 - 442 \text{ kW/cm}^2$ (by assuming that the 200 fs excitation pulse is spread out over the 1 ns time window of the PL-decay). In combination with the 4K measurement temperature, these high excitation fluences are further saturating the nonradiative recombination centers at the measurement conditions, thus establishing the radiative limit more firmly.

Hex-SiGe seems to outperform many III/V semiconductors in terms of its radiative efficiency at elevated temperatures¹. This was particularly evident in Fig. 4 of the paper of Fadaly et al. (Nature, 580, 205–209, 2020), where it was shown that the radiative efficiency was the same at 300K as at 4K. One of the reasons for the high radiative efficiency is the fact that the dominant I₃ defect does not induce a state within the bandgap^{2,3}. Moreover, surface recombination in hex-Ge and hex-SiGe is found to be quite weak⁴. The combination of the weak surface recombination and the absence of nonradiative recombination centers due to the I₃ defect at least partially explain the high internal radiative efficiency of hex-SiGe.

In addition to the extensive evidence for direct bandgap emission of hex-SiGe in our previous paper (Fadaly et al., Nature, 580, 205–209, 2020), we recently obtained a radiative lifetime of 1.7 ns for bulk hex-Ge at a temperature of 4K (much shorter than the 20 μ s predicted by theory) by both picosecond pump-probe reflectivity measurements at 4K, as well as by a Lasher-Stern-Würfel analyses of the PL-lineshape. The reviewer correctly states that such a fast lifetime might also be due to nonradiative recombination. The PL lineshape analysis however provides an independent method to obtain the oscillator strength of hex-Ge, which is $f \geq 8.5$ (compared to $f = 17$ for ZB GaAs and $f \geq 5.2$ for WZ GaN), thus showing that the 1.7 ns lifetime in hex-Ge is due to interband radiative recombination. Hex-Ge is an elemental semiconductor being the endpoint of the hex-SiGe family of alloys. Until recently, it was believed that radiative recombination in hex-SiGe is efficient due to the breaking of the translational symmetry due to alloy disorder (see Fadaly Fig. 1), but hex-Ge has no alloy disorder and was expected to suffer from a very small oscillator strength of $f = 0.02$. Since we now have convincing experimental evidence that hex-Ge also features a high oscillator strength in combination with efficient carrier recombination, it is very likely that this applies for all direct bandgap hex-SiGe alloys. We will add a reference to this manuscript/paper in the final text.

The observation of a LILO slope below unity is not an indication for nonradiative recombination. Theoretically, we expect a slope of 1 for pure radiative recombination, a slope of 2 for pure Shockley-Read-Hall nonradiative recombination and a slope of 2/3 for Auger recombination. For bulk hex-SiGe we observe slopes between 0.5 and 1, where the lower slopes are most probably correlated with a high density of stacking faults in the core or I_3 defects in the shell, both of which lead to cubic insertions into our hex-SiGe nanowire shells. At increasing excitation density, more and more carriers are injected into these cubic insertions where they are temporarily lost for efficient radiative recombination. In our quantum well samples, carriers might also overflow into the barrier. These processes might explain why our LILO slopes are slightly below unity.

We added the discussion about the radiative limit in the main text between line 299 and 325. The explanation for the LILO slopes is added from line 221 to line 233.

2) Intermixing was not taken into account for the QWs. For the used high growth temperatures of $\sim 600^\circ\text{C}$, significant Ge surface segregation has to be expected (which might be seen in Fig. 2(c) in the form of very blurry interfaces. This would change the QW shape and shift the emission to higher energies as compared to the theoretical results (Fig. 3(b)).

We agree with the reviewer that interdiffusion is a possible mechanism to explain the Si content in the Ge wells. In response to this reviewer, we added this as an explanation of the shift of the PL emission energies to higher energy in the main text, line 452-459.

3) Doping was not taken into account. This is somewhat surprising since the authors have stated in their earlier Nature paper (Fadaly et al.) that the presence of the GaAs core dopes the hex-group-IV to $\sim 10^{19}\text{cm}^{-3}$. This will likely influence the optoelectronic behavior of the transitions. Please indicate why this can be neglected.

The hex-SiGe in an earlier Nature paper (Fadaly et al.) was indeed heavily doped with arsenic. We made progress in reducing the As doping by introducing coating runs between SiGe growth runs to coat the inner parts of the reactor with a SiGe capping layer to reduce As-evaporation. This growth procedure reduced the As doping concentration below the detection limit of Atom Probe Tomography, so $\text{As} < 2.5 \times 10^{18}\text{cm}^{-3}$. We do believe the main dopant in the shells is still As. The

reduction in doping concentration, compared to previously published results, is now mentioned in the main text in line 91-93 and the underlying reasons are discussed in the methods section of the extended data at the bottom of page 1 and the top of page 3.

Minor 4) The mix of providing the layer thickness in nm and minutes is confusing and disturbs the reading flow. Please correct to nm throughout.

We now consistently refer to the thickness in nm, except for the first time we refer to the thickness of a sample, where we give both the growth time and the thickness in nm.

Minor 5) In the abstract, the authors state that Si is a poor choice for active photonic applications. Modulators and Detectors are active and commercially available. Only emission is a problem. The authors might want to change that.

We thank the reviewer for this comment. This is an obvious mistake made by us, and the reviewer is correct. The abstract has been changed from: "Si is a poor choice for active photonic applications", towards: "Si is a poor choice as a light emitter for photonic applications". The updated line better reflects the state-of-the-art, and it is indeed what we envisioned to say in the first place.

Reviewer #3 (Remarks to the Author):

Referee report on the paper of W.H.J. Peeters et. al. "Direct bandgap quantum wells in hexagonal Silicon Germanium"

The current manuscript represents the development of the previously published idea (see, for example, Nature 580, 205 (2020)) devoted to the Ge and SiGe band structure modification via creating the hexagonal form of such materials. According to the previous results, Ge and SiGe in the hexagonal arrangement should be the direct-gap semiconductors and this could help to overcome the main problem of their inherently low light emission efficiency and so potentially enable the construction of Si-compatible light source using group-IV materials. In particular, in the current manuscript it is proposed to use somewhat "classical" approach of double heterostructure - the Ge QW (or SiGe QW with low Si fraction) as an active medium that is sandwiched between SiGe barrier layers.

In the current manuscript thorough theoretical investigation of bandgap modification of hexagonal Ge/SiGe structures was presented and rather large amount of experimental data was obtained. Analysis of the hex-Ge/SiGe photoluminescence (PL), in particular, dependencies of the PL peak position on the hex-Ge thickness and on the excitation density has proven the realization of type-I heterojunction in which both electrons and holes are confined in the hex-Ge QW region. The latter is of indisputable importance.

We thank the reviewer for these comments.

However, I have significant objections related to the goal setting of this work, as it was stated that it is aiming "for integrating light emitting materials on silicon". Indeed, all studied structures were fabricated on GaAs(111)B substrates, but not on silicon, and so usage of such substrate type and orientation would allow only the heterogeneous integration with Si or SOI platforms. As such, the

fabrication procedure of hex-Ge/SiGe NW-based structures, presented in the manuscript, does not seem any easier (if it's heavier) than integration of “conventional” planar A3B5 structures on silicon. To the contrary, both heterogeneous and hybrid or even monolithic integration of A3B5 QW-and QD-based light-emitting structures on silicon was very thoroughly studied and now they are already used in real optoelectronics devices. It seems questionable that the GaAs NW-based hex-Ge structures would be easier and cheaper (!) to integrate with silicon as compared to the former “conventional” A3B5-based ones.

Moreover, as regards to group-IV materials, it seems that usage of strained-Ge or GeSn structures which could be directly fabricated on silicon would be more promising approach as compared to hex-Ge NW-based ones. Both strained-Ge and GeSn systems has already shown the laser action at cryogenic and even at room temperature (see, for example, Nat. Commun. 10:2724 (2019); Nat. Phot. 14, 375 (2020); Optica 7, 924 (2020); APL 120, 051107 (2022)). Electrical pumping which is needed in real devices was also demonstrated.

Furthermore, I have not found any significant advances in light-emitting properties of hex-Ge/SiGe structures, reported in this manuscript, compared to the previous work published in Nature 580, 205 (2020).

We fully agree that to be competitive with the present work on the hybrid integration of III/V semiconductor lasers on Si-photonics by e.g. the Bowers group, we need to fabricate hex-SiGe in a Si compatible process. We are presently investigating multiple integration approaches to achieve this goal, including the transformation of cubic SiGe nanofins into hex-SiGe, which is a CMOS compatible approach pioneered by IMEC¹⁰. Other approaches include the growth of planar hex-SiGe on approximately lattice matched substrate, followed by the transfer to an SOI substrate. In the meantime, it is important to investigate the potential of this new material system e.g. the lasing and single photon emission properties.

We admire the work on GeSn which is well-known to us. That work is scientifically very interesting, but it is also still far from real-world applications when looking to e.g. the presently achieved threshold current densities at room temperature. We also note that the first publications on GeSn showing a direct bandgap are 24 years old¹¹, while we first reported direct bandgap hex-SiGe in 2020. The work on GeSn is thus about two decades ahead of that on hex-SiGe, and only the future will be able to tell us which material platform is eventually going to survive towards a new technology. The advantages of hex-SiGe for photonics are the achievement of direct bandgap emission without the need for strain, as well as very high radiative efficiency¹² at 300K. Our recent observation of stimulated emission and optical gain of at least 545 cm^{-1} , which has recently been submitted, also constitutes a major advancement for this material.

Hex-SiGe is not only a promising semiconductor for photonics, we think it will also find applications in quantum information. Qubits in hex-SiGe are promising since hex-SiGe can be fabricated from nuclear spin free isotopes. Moreover hex-SiGe features a large g -factor of $g=18$. Like all other direct bandgap group IV semiconductors, it has no valley degeneracy in contrast to the cubic SiGe system. Last but not least, hex-SiGe has a weak electron-optical phonon coupling due to the absence of the Fröhlich polar electron-optical phonon coupling. These factors should enable fast spin-based qubits with long coherence times combined with fast qubit manipulation. The type I band alignment of hexagonal Ge QWs embedded in SiGe barriers allows for spin qubits based on electrons as well as holes.

We have added a paragraph in the introduction of the main text on hex-SiGe qubits, see line 45-50 as well as a sentence on future planar hex-SiGe QWs in the conclusions, line 483-485.

I have also several smaller remarks:

1. Some differences between the hex-Ge/SiGe structures used for theoretical calculations and for experimental examination exist. Band structure was calculated for MQW hex-Ge/SiGe structures having thin SiGe barriers and thicker Ge QWs, while single-Ge QW structures embedded into thicker SiGe barriers were experimentally fabricated and studied. This could potentially mislead the readers.

We thank the reviewer for this remark, which helped us to decide to move the theory part towards the end of the paper. The experimental barrier thickness is always larger than 50 nm, see main text line 110, but the geometry is also clear from e.g. Fig. 2a.

Indeed, the ab initio modelling uses a Multiple Quantum Well structure, a superlattice, i.e. an artificial translational symmetry in the growth direction, as typical in all plane-wave-like descriptions. For thick-enough barrier layers the results for lowest $n=1$ subbands should however not be different from those for isolated QWs. We have tested that the used small thickness of about 2 nm is sufficient to avoid overlapping of the exponentially decaying wave functions of at least the lowest confined state in the considered QWs. A possible influence due to biaxial strain induced by pseudomorphic growth of the SiGe barrier and Ge well materials remains negligible, since the significant strain measured in the actual thick barriers appearing in the experimentally studied heterostructures is applied as “external” strain in the ab initio calculations.

These arguments for the correct modelling are now described in more detail in the text (see lines 349-356) of the resubmitted modified manuscript.

2. It is not clear from theoretical calculations why the Ge layer thickness affects the lowest conduction band of the bulk $\text{Si}_{0.25}\text{Ge}_{0.75}$ barrier material (red solid line in Fig. 1d (now Fig. 6d)). It is also unclear what is the reason for the different behavior of the calculated values for electron and hole band offsets of Ge/ $\text{Si}_{0.25}\text{Ge}_{0.75}$ structures (Fig. E2c (Now Fig. E10c) in extended data).

The structural optimization within the DFT approach of the studied MQW structures gives rise to mutual biaxial strains in the hex-Ge well layers as well as in the SiGe barrier layers in dependence of the layer thicknesses in addition to the significant “external” biaxial strain taken from the measurements. Despite this strong biaxial strain due to the assumed pseudomorphic growth of the hex-Ge/SiGe heterosystems on the wurtzite-GaAs core wires, the additional small strain distribution in the heterosystem only slightly affects the actual strong strain situation in the barrier material resulting in small band edge variations made visible in Fig. 6d of the revised manuscript (Fig. 1d in the old manuscript) by a red (blue) line for electrons (holes). The accompanying changes of the QW barrier heights of less than 0.015 eV hardly influence the carrier confinement in the lowest $n=1$ levels in the QWs.

These facts are also mentioned in the resubmitted “Theoretical and Numerical methods” section in the Extended Data.

3. From the reported results it seems that lattice constants of NW hex-Ge/ $\text{Si}_{0.2}\text{Ge}_{0.8}$ are close to lattice constants of bulk hex- $\text{Si}_{0.2}\text{Ge}_{0.8}$, and strain relaxation in Ge layer does not generate defects (Fig. E5 in extended data). But how the strain relaxation between GaAs NW and hex- $\text{Si}_{0.2}\text{Ge}_{0.8}$ shell occurs?

Indeed, there is a lattice mismatch between the Ge and $\text{Si}_{0.2}\text{Ge}_{0.8}$, which seems to be relaxed without a misfit dislocation. As the reviewer points out, there is also a lattice mismatch between

the GaAs and SiGe barrier. Here, the lattice mismatch is predominantly relaxed through elastic relaxation, in which the GaAs core is compressed due to the overgrowth of $\text{Si}_{0.2}\text{Ge}_{0.8}$ shells. The total interplay between all layers results in the compression of the Ge QWs of -0.6% and -0.91 % along the [11-20] and [0001] directions, as discussed in the main text in lines 151-158 and 360-364.

Apart from elastic relaxation, we also note the presence of some dislocation boundaries in the shells, as discussed in the main text in lines 279-289. These dislocation boundaries occur more frequently in the samples with a higher Si content in the barriers. Thus, we concluded that lattice-mismatch could be a cause of these dislocations.

4. Large spread of hex-Ge QW thickness in a single NW and cubic phase inclusions make the PL interpretation questionable. According to the Fig. 3b in main text and Fig. E3b (Now Fig 2b in the main text) in extended data, the Ge layer after 9 min of growth possesses larger thickness deviation and smaller average thickness, than the similar Ge layer after 6 min of growth. At the same time, the PL peak related to the structure with 9 min of Ge growth has smaller width and is blue shifted as compared to PL peak related to the structure with 6 min Ge growth. I think, there is a contradiction here.

We emphasize that, if one properly takes the error bars into account, one cannot claim that the 6 min QW sample is blue shifted with respect to the 9 min sample. The error margins are of importance since we might have measured the QW thickness with TEM from a different subset of the nanowires probed in our PL-setup.

Moreover, we acknowledge that the growth kinetics are not fully understood, as this is a new material system. We found that the growth rate of the QW layer is correlated with the thickness of the initial SiGe barrier. The thickness the SiGe barrier of the 9 min (6 min) QW samples turned to be thinner (thicker) than expected, resulting in in thinner (thicker) QWs.

5. It seems that low temperature PL spectra presented for 9 min Ge growth structure in Fig.4c (Now Fig. 4e) possess the pronounced double-hump view, while the PL spectrum for the same structure presented in Fig.3a (red line) is much more like a single-hump one. Please comment. Also, the PL temperature quenching for different structures should be discussed in more detail.

We completely agree with the reviewer that the detailed behavior of the double-peak structure in the old Fig. 4c, which is the new Fig. 3d,e, is interesting and deserves a more detailed investigation. We revised the manuscript by adding Figs. 4b and 4d to show the reader the full dataset obtained for both a thin QW Fig. 4a-c) and a thick QW (Fig. 4d-f). In addition, we shortly discuss the main features in Figs. 4d,e, which are due (i) to Burstein-Moss bandfilling at increasing excitation, (ii) the possible observation of the CB2-HH2 second confinement level transition at low temperature and high excitation, (iii) carrier localization in the hex-SiGe barrier at low temperature and a more efficient carrier diffusion towards the QW at increasing temperature, and (iv) the spectral diffusion due to carrier diffusion from the narrow QWs to the thicker QWs at elevated temperature. A more detailed discussion is beyond the scope of the present paper.

We thank the reviewer for his suggestion to investigate the temperature quenching. The analysis of the thermal quenching now yields experimental data for the value of the band offset. The analysis is presented as a new Figure panel: Fig. E6d. In addition, we added the following paragraph to the main text (lines 257-265): "From the thermal quenching results, we estimate the band offset and the effective mass of the most shallow confined charge carrier of the three widest

(approximately infinite) QW samples are found to be $E_{\text{offset}} = (100 \pm 30) \text{ meV}$ and $m^* = (0.03 \pm 0.02) m_0$ respectively, which is close to the predicted band offset and effective mass by our ab-initio bandstructure calculations which are presented below”.

6. The studied structures have various defects (cubic phase inclusions, twins etc) as reported by authors. So it seems questionable that the measured PL decay time is determined precisely by the radiative, and not the non-radiative processes.

See comment 1 reviewer 2. We have strong arguments that the emission is purely radiative at low temperature. We added this discussion in the main text between line 299 and 321.

References:

1. E.M.T. Fadaly et al. Direct Band Gap Emission from Hexagonal Ge and SiGe Alloys. *Nature* **580**, 205–209 (2020).
2. Fadaly, E. M. T. et al. Unveiling Planar Defects in Hexagonal Group IV Materials. *Nano Letters* **21**, 3619–3625 (2021).
3. Vincent, L. et al. Growth-Related Formation Mechanism of I3-Type Basal Stacking Fault in Epitaxially Grown Hexagonal Ge-2H. *Advanced Materials Interfaces* **9**, 2102340 (2022).
4. Berghuis, W. J. H. W. J. et al. Low Surface Recombination in Hexagonal SiGe Alloy Nanowires: Implications for SiGe-Based Nanolasers. *ACS Applied Nano Materials* **7**, 2343–2351 (2024).
5. Leroux, M. et al. Temperature quenching of photoluminescence intensities in undoped and doped GaN. *Journal of Applied Physics* **86**, 3721 (1999).
6. G. Bacher, H. Schweizer, J. Kovac, and A. F., Bacher, G., Schweizer, H., Kovac, J. & Forchel, A. Influence of barrier height on carrier lifetime in GaAs/GaAs single quantum wells. *Phys. Rev. B* **43**, 9312–9315 (1992).
7. Lourenço, S. A. et al. Effect of temperature on the optical properties of GaAsSbN/GaAs single quantum wells grown by molecular-beam epitaxy. *Journal of Applied Physics* **93**, 4475–4479 (2003).
8. Schenk, H. P. D., Leroux, M. & De Mierry, P. Luminescence and absorption in InGaN epitaxial layers and the van Roosbroeck-Shockley relation. *Journal of Applied Physics* **88**, 1525–1534 (2000).
9. Lambkin, J. D. et al. Temperature dependence of the photoluminescence intensity of ordered and disordered In_{0.48}Ga_{0.52}P. *Appl. Phys. Lett.* **65**, 73–75 (1994).
10. Qiu, Y. et al. Epitaxial diamond-hexagonal silicon nano-ribbon growth on (001) silicon. *Scientific Reports* **5**, 1–10 (2015).
11. Ragan, R. & Atwater, H. A. Measurement of the direct energy gap of coherently strained. *Nature* **3420**, 3418–3420 (2000).
12. Fadaly, E. M. T. et al. Direct-bandgap emission from hexagonal Ge and SiGe alloys. *Nature* **580**, 205–209 (2020).

REVIEWER COMMENTS

Reviewer #1 (Remarks to the Author):

The authors addressed most of the points raised by the referees and the manuscript has been improved substantially, thus it could be accepted for publication in Nature Communications. Still, I have a couple of minor comments and suggestions regarding the authors' answers (ref. 1).

Q2. The resulting strain calculated along the [1-100] direction should be added after line 366

Q5 and Q8. I appreciate the authors's answers and the tentative to improve the manuscript text regarding these points. Still, I believe that using different colours for the points in Fig. 2b for each of the 6 facets would improve further the manuscript.

Reviewer #2 (Remarks to the Author):

The authors did a great job in their very detailed answers to the referees. I have still one issue with their reply and updated version of the paper and it concerns their statement on the direct band gap.

The authors wrote:

"Semiconductors generally show almost pure radiative recombination at low enough temperature.

The PL-intensity plotted in Figs. 4f and E6c, is proportional to the internal quantum efficiency for

radiative emission, which varies with temperature according to the ratio of the radiative recombination rate, divided by the total recombination rate

This (and the following) is a reasonable argument but it intrinsically implies that the material is seen as direct bandgap material without taking into account that it could be indirect. The first sentence of the statement is not true, as it only accounts for direct bandgap semiconductors. Nobody would assume that Si at very low excitation powers and $T = 4K$ would emit like InGaAs.

The authors (Absolute experts in nanowire growth) could clear the doubts whether this material is direct or not once and for all, if they would compare the material to, say InAs nanowires of similar sheet densities. They emit at similar wavelengths, and, given all the points the authors elaborated on, I would expect that emission intensities would match within about an order of magnitude or so if the material would be direct. If it is not direct the differences between hex SiGe and InAs should be much higher, like $>10^3$.

I fully understand that the authors could hesitate to incorporate a such a comparison in this work. If they decide to, the manuscript and the main point - a direct band gap in hex SiGe QWs - would be much stronger. If not, the paper is still very good, but it could leave doubts [especially, since the theoretical and experimental lifetimes don't match by four! orders of magnitude, as the authors stated].

Reviewer #3 (Remarks to the Author):

Authors have carried out the substantial work to improve their manuscript. In particular they re-arranged the manuscript which made the text flow more convenient, revised some figures and added new data and discussion. Authors provided detailed answers for my questions and remarks and modified the text accordingly. They also mentioned in the response letter that they have already obtained even more interesting data, concerning the realization of stimulated emission in such kind of structures. The latter is another major step to achieve the group-IV based laser which would be very interesting to see.

I have only two minor remarks.

1. Line 39-41. It is stated that hex-SiGe emission was observed down to 1.5 μm . However, I have not found the experimental spectra with such a wavelength neither in the author's previous paper (Fadaly et.al., Nature 508, 205 (2020)) nor in the present manuscript even at low temperatures. Bandgap shrinkage with increasing temperature will complicate the task to decrease the emission wavelength of hex-SiGe even further. Either related references should be provided where the emission at 1.5 μm was experimentally demonstrated or the wavelength range in the manuscript text should be corrected.

2. It would be more clear for readers if authors provide the energy scale in Fig.1b where the band alignment in hex-Ge NW is schematically shown.

Finally, I think that the manuscript could be accepted after making the above-mentioned minor corrections (without the need for additional review).

REVIEWER COMMENTS

Reviewer #1 (Remarks to the Author):

The authors addressed most of the points raised by the referees and the manuscript has been improved substantially, thus it could be accepted for publication in Nature Communications. Still, I have a couple of minor comments and suggestions regarding the authors' answers (ref. 1).

Q2. The resulting strain calculated along the [1-100] direction should be added after line 366
The reviewer is correct that the value of the relaxation was missing. We added in line 366 the following text:

"The studied heterostructure is allowed to relax along the [1100] direction, resulting in an 0.3% expansion in the well and 0.1% contraction in the $Si_{0.25}Ge_{0.75}$ barrier."

Q5 and Q8. I appreciate the authors's answers and the tentative to improve the manuscript text regarding these points. Still, I believe that using different colours for the points in Fig. 2b for each of the 6 facets would improve further the manuscript.

Indeed, the QWs are grown on 6 facets of the NW. These six facets all belong to the same crystal family, the {1-100} planes. Thus, the QW thickness variations are not due to a growth rate that depends on the crystal direction.

We attempted to implement the suggestion of the reviewer, by assigning a different color to each facet, after ranking the QW thicknesses in descending order. An example of one of the QWs is shown below in Fig. 1.

In practice, however, these details get lost in the graph when the same method is applied to all 7 QW samples. The visualization introduces many colours in the graph which makes it more difficult to interpret. Moreover, it still does not capture the full picture, as the thickness of the neighbouring facets determines if the charge carriers can diffuse toward the thickest QW. Hence, we opted to keep the same colour for each of the QW thicknesses of each sample Fig. 2b.

Figure 1: Thickness of the Ge QW, for the (12 ± 3) nm (2.0 min) sample. All 6 facets from three different NWs are measured. The QW thickness for each NW is ranked from largest to thinnest, according to the legend. The coloured area shows the approximate probability distribution, obtained by Kernel smoothing

Reviewer #2 (Remarks to the Author):

The authors did a great job in their very detailed answers to the referees. I have still one issue with their reply and updated version of the paper and it concerns their statement on the direct band gap.

The authors wrote:

"Semiconductors generally show almost pure radiative recombination at low enough temperature. The PL-intensity plotted in Figs. 4f and E6c, is proportional to the internal quantum efficiency for radiative emission, which varies with temperature according to the ratio of the radiative recombination rate, divided by the total recombination rate

This (and the following) is a reasonable argument but it intrinsically implies that the material is seen as direct bandgap material without taking into account that it could be indirect. The first sentence of the statement is not true, as it only accounts for direct bandgap semiconductors. Nobody would assume that Si at very low excitation powers and $T = 4\text{K}$ would emit like InGaAs.

The authors (Absolute experts in nanowire growth) could clear the doubts whether this material is direct or not once and for all, if they would compare the material to, say InAs nanowires of similar sheet densities. They emit at similar wavelengths, and, given all the points the authors elaborated on, I would expect that emission intensities would match within about an order of magnitude or so if the material would be direct. If it is not direct the differences between hex SiGe and InAs should be much higher, like $>10^3$.

I fully understand that the authors could hesitate to incorporate a such a comparison in this work. If they decide to, the manuscript and the main point - a direct band gap in hex SiGe QWs - would be much stronger. If not, the paper is still very good, but it could leave doubts [especially, since the theoretical and experimental lifetimes don't match by four! orders of magnitude, as the authors stated].

We thank the reviewer for raising his concerns. We apologize that we did not specify that the first sentence of our first rebuttal only applies to direct bandgap semiconductors. We already compared the external radiative efficiency of our bulk hex-Si_{0.2}Ge_{0.8} nanowires with a high-quality InGaAs/InP multiple quantum well (MQW) samples in the Methods section called “External radiative efficiency of hex-SiGe” and Extended Data Fig. 10 of our paper “Fadaly et.al., Nature 508, 205 (2020)”, yielding an external radiative efficiency of 2%. We recently repeated this analysis, yielding the same result. Apart from this fact, our Nature paper showed a wealth of evidence for direct bandgap radiative recombination at 4K, including a light-in-light-out slope of 1.02 (inset Fig. 4e in the Fadaly paper) as well as an almost equal amount of radiative recombination at 4K and at 300 K, as again shown in Fig. 4e of that paper. In the present manuscript, we obtain additional evidence for radiative recombination at 4K in Fig. 5d of the main text, where we observe a decreasing lifetime as a function of excitation, which is expected for the radiative recombination mechanism $\frac{1}{\tau_{rad}} = B(n_0 + \Delta n)(p_0 + \Delta p)/\Delta p$, in which n_0 and p_0 are the doping concentrations and $\Delta n, \Delta p$ are the photoexcited carrier concentrations. For large excitation ($\Delta n = \Delta p$, $\Delta n > n_0$, $\Delta p > p_0$), the expression for the radiative lifetime becomes $\frac{1}{\tau_{rad}} = B\Delta n$, showing a decreasing lifetime τ_{rad} for increasing excitation densities $\Delta n = \Delta p$. On the contrary, for nonradiative recombination, we expect that nonradiative recombination centres get saturates at increasing excitation density, yielding an increasing recombination lifetime at increasing excitation, contrary to our experimental results in Fig. 5d.

We emphasize that our claim for a direct bandgap involves the alignment of the Γ -point with respect to the \bar{M} - point and not the external radiative efficiency, which is presently not fully understood. The combination of our theoretical calculations and the argument we already provided in the main text (Fig. 4f), we have a large amount of evidence for a direct bandgap, which is a main conclusion from both the Nature paper (Fadaly et al.) and the present paper.

We added the following paragraph to the main text line 325-340:

“We note that the observation of a decreasing recombination lifetime with increasing excitation density provides additional evidence for radiative recombination governed by $1/\tau_{rad} = B(n_0 + \Delta n)(p_0 + \Delta p)/\Delta p \approx B\Delta n$ for high excitation ($\Delta n = \Delta p \gg n_0, p_0$), in which B is the coefficient for radiative recombination, n_0, p_0 are the doping concentrations and $\Delta n, \Delta p$ are the photoexcited carrier concentrations. On the other hand, the observations in Fig. 5d cannot be explained by a nonradiative recombination mechanism since nonradiative recombination centers get saturated at high excitation, thus increasing the lifetime. We conclude that the observed nanosecond radiative recombination lifetime falls within the same range as that reported by Fadaly et. al. [19, 46] for bulk hex-SiGe nanowires and confirms direct bandgap emission in Si_{0.1}Ge_{0.9}/Si_{0.3}Ge_{0.7} QWs.”

Reviewer #3 (Remarks to the Author):

Authors have carried out the substantial work to improve their manuscript. In particular they re-arranged the manuscript which made the text flow more convenient, revised some figures and added new data and discussion. Authors provided detailed answers for my questions and remarks and modified the text accordingly. They also mentioned in the response letter that they have already obtained even more interesting data, concerning the realization of stimulated emission in such kind of structures. The latter is another major step to achieve the group-IV based laser which would be very interesting to see.

I have only two minor remarks.

1. Line 39-41. It is stated that hex-SiGe emission was observed down to 1.5 μm . However, I have not found the experimental spectra with such a wavelength neither in the author’s previous paper (Fadaly et.al., Nature 508, 205 (2020)) nor in the present manuscript even at low temperatures. Bandgap shrinkage with increasing temperature will complicate the task to decrease the emission

wavelength of hex-SiGe even further. Either related references should be provided where the emission at 1.5 μm was experimentally demonstrated or the wavelength range in the manuscript text should be corrected.

We again like to thank the reviewer for critically reading of the manuscript. In the Nature paper of 2020, we were not able to measure the PL-spectrum of a single hex-SiGe nanowire. Using the improved setup as described in the methods section, part E, we are currently able to measure PL-spectra from single nanowires, showing emission down to 1.5 μm . The wavelength tunability for bulk hex-Si_xGe_{1-x} is shown in Fig. 2 below, where we added micro-PL spectra for hex-Si_{0.5}Ge_{0.5} and hex-Si_{0.61}Ge_{0.39} to the spectra of the 65%-100% Ge nanowires as already published in Nature 508, 205 (2020). The two additional spectra are measured at a high excitation density of 0.6 mJ/cm², thus highlighting band-to-band transitions while saturating impurity dominated transitions. The observation of photoluminescence down to 1.5 μm , plotted on a linear intensity scale, is strongly suggesting that it is feasible to observe direct bandgap emission down to 1.5 μm . But as already stated in the main text, “the limits of which (the tunability region below 2.0 μm) are a subject of future investigations”.

Figure 2: Tunability of the photoluminescence of hex-Si_xGe_{1-x} for Ge compositions between 100% and 39%. It should be noted that the samples with Ge-compositions between 100% and 77% were measured in macro-PL, the 60% and 65% samples were measured on bunches of transferred hex-Si_xGe_{1-x} nanowires. The grey curve represents the hex-Si_{0.5}Ge_{0.5} sample and the black curve is the hex-Si_{0.61}Ge_{0.39} sample which were measured with micro-PL at an excitation density of 0.6 mJ/cm². The micro-PL experiments are further explained in the Methods section.

2. It would be more clear for readers if authors provide the energy scale in Fig.1b where the band alignment in hex-Ge NW is schematically shown.

We improved Fig. 1b to include the band offsets and bandgap, as suggested by the reviewer.

Finally, I think that the manuscript could be accepted after making the above-mentioned minor corrections (without the need for additional review).

REVIEWERS' COMMENTS

Reviewer #2 (Remarks to the Author):

The authors implemented all suggested changes. The manuscript reads very well, the research is timely and of impact and I suggest to publish as is.